# Norway spruce postglacial recolonization of Fennoscandia

Kevin Nota [1], Jonatan Klaminder [2], Pascal Milesi[1,3], Richard Bindler [2], Alessandro Nobile[1], Tamara van Steijn[1,2], Stefan Bertilsson [1,4], Brita Svensson[1], Shun K. Hirota [5], Ayumi Matsuo[5], Urban Gunnarsson [6], Heikki Seppä [7], Minna M. Väliranta [8], Barbara Wohlfarth[9], Yoshihisa Suyama [5] & Laura Parducci [1,10 ✉]

Contrasting theories exist regarding how Norway spruce (*Picea abies*) recolonized Fennoscandia after the last glaciation and both early Holocene establishments from western microrefugia and late Holocene colonization from the east have been postulated. Here, we show that Norway spruce was present in southern Fennoscandia as early as 14.7 ± 0.1 cal. kyr BP and that the millennia-old clonal spruce trees present today in central Sweden likely arrived with an early Holocene migration from the east. Our findings are based on ancient sedimentary DNA from multiple European sites ($N = 15$) combined with nuclear and mitochondrial DNA analysis of ancient clonal ($N = 135$) and contemporary spruce forest trees ($N = 129$) from central Sweden. Our other findings imply that Norway spruce was present shortly after deglaciation at the margins of the Scandinavian Ice Sheet, and support previously disputed finds of pollen in southern Sweden claiming spruce establishment during the Lateglacial.

[1] Department of Ecology and Genetics, Evolutionary Biology Centre, Uppsala University, Norbyvägen 18D, SE-75236 Uppsala, Sweden. [2] Department of Ecology and Environmental Science, Umeå University, Umeå, Sweden. [3] Scilifelab, Uppsala, Sweden. [4] Department of Aquatic Sciences and Assessment, Swedish University of Agricultural Sciences, Uppsala, Sweden. [5] Field Science Center, Graduate School of Agricultural Science, Tohoku University, 232-3 Yomogida, Naruko-onsen, Osaki, Miyagi 989-6711, Japan. [6] Swedish Environmental Protection Agency, SE-106 48 Stockholm, Sweden. [7] Department of Geosciences and Geography, University of Helsinki, Helsinki, Finland. [8] Environmental Change Research Unit (ECRU), Ecosystems, Environment Research Programme, University of Helsinki, Helsinki, Finland. [9] Department of Geological Sciences, Stockholm University, and Bolin Centre for Climate Research, SE-10691 Stockholm, Sweden. [10] Department of Environmental Biology, Sapienza University of Rome, Piazzale Aldo Moro 5, 00185 Rome, Italy. ✉email: laura.parducci@uniroma1.it

Norway spruce (*Picea abies* (L.) H. Karst) is one of the most socioeconomically important tree species in northern Europe, yet its postglacial colonization history in Fennoscandia remains clouded in uncertainty with contrasting migration theories[1–4]. The distribution range in Europe is divided into a vast northern domain that covers Fennoscandia, Belarus, the Baltic states, and European Russia and a southern domain that covers parts of central Europe, the Alps, and the Carpathians. Populations from the two domains are genetically differentiated and their divergence is ancient (15 mya)[5]. Pollen records from the northern domain suggest a late Holocene migration event where spruce arrived in central Sweden 2000-3000 calibrated years before present (cal. yr BP) from the east via a north-eastern route through Finland[6,7]. Other pollen studies show a much earlier presence of spruce in southern Sweden[8] and immediately east of the Scandinavian Ice Sheet (SIS)[9,10] during the Lateglacial (15-11.7 cal. kyr BP), but in quantities too low to be considered of local origin. Segerström and Stedingk, however, suggested that in some settings low pollen counts of spruce can be considered of local origin[11]. Findings of spruce megafossils dating back to glacial times[12–15], as well as ancient DNA fragments of *P. abies* in early Holocene sediments seem to support the theory that spruce was present in the central Scandinavian Mountains (the Scandes) already at 13 cal. kyr BP[2,13]. Yet, this interpretation has been challenged because of the limited number of pollen grains found in mid-to-late-Holocene sediment records in the region[16]. Moreover, we lack a good explanatory model for spruce migration during the glacial to early Holocene transition in Fennoscandia.

Genomic data from contemporary spruce forests rule out the possibility that southern European refugia (Alps and Carpathian) contributed to the recolonization of Fennoscandia[5,17,18]. It appears, however, that both theories of an eastern and a western origin of spruce establishment directly after the deglaciation in central Fennoscandia is problematic. A migration route from the east means that spruce rapidly dispersed across a massive ice sheet that persisted in northern Fennoscandia until ca 10 cal. kyr BP[19,20] or that spruce established from the south-east and moved toward the north to central Sweden, for which the palynological record is controversial[8,21,22]. The alternative hypothesis of an early colonization from the west where spruce survived on summits protruding the ice-sheet (the nunatak hypothesis) was first suggested based on early Holocene macrofossil findings of spruce in the alpine region of the southern Swedish Scandes[13]. Later, this hypothesis was supported by the discovery of a *P. abies* mitochondrial DNA mutation (mh05 haplotype A) dominant in contemporary forests from western Fennoscandia and present in ancient sediments from central Norway dated to 10.3 cal. kyr BP[2]. A limited persistence of spruce in ice-free glacial microrefugia in the west has been strongly questioned[4,16,23], based on the limited support found in pollen records and the lack of suitable habitats for spruce. The authors concluded that 'spruce persistence in this region was against all ecological expectations'[4]. However, the strongest evidence of early Holocene establishment of spruce trees in central Fennoscandia comes from still living and clonally regenerating spruce trees currently growing in the central Scandes several metres above the treeline, and where carbon-14 inferred ages suggest an onset of growth around 9.7–9.5 cal. kyr BP[14]. Each of these clones consists of several tightly growing trees with or without a trunk, rarely higher than a few meters and likely belonging to a single ancient individual (Fig. 1A, B). It has been suggested that these clones originated either from local cryptic refugia, or from early colonizing trees surviving on the west coast of Norway[12–14,24], but origins from the east[25] or from the south cannot be excluded. Mitochondrial and nuclear DNA from contemporary spruce forests indeed suggest a complex spread

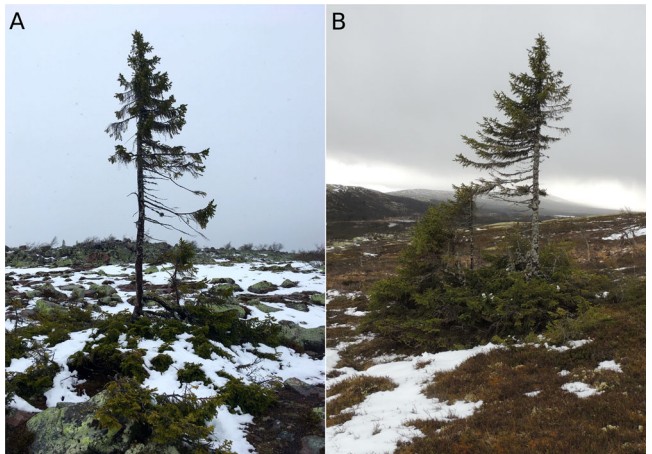

**Fig. 1 Clonal spruce trees from Dalarna County in Sweden. A** Old Tjikko spruce with the oldest fossil remains dated 9.5 cal. kyr BP. **B** Gunnar Samuelsson's spruce (GS-spruce) with fossil remains dated 6.3 cal. kyr BP.

into Fennoscandia from several areas[2,17,26], while the direction of spread reconstructed from pollen alone hardly explains these patterns. A combined pollen and DNA analysis indicates that Norway spruce spread into Fennoscandia long before it became abundant, and that this spread occurred without leaving a clear trace in the palynological record[25]. It is possible that this first spread from the east occurred via seeds transported over kilometres on ice-crusted snow[6] giving rise to small, isolated outpost populations, which survived for thousands of years in the central Scandes without contributing to the gene pool of later colonising trees[27] and which were too small to be detectable by conventional pollen analysis. Giesecke (2013)[25] concluded that 'studies on ancient DNA from several sites are needed to obtain more evidence for enhancing our understanding of the postglacial history of this important tree species in Scandinavia'[25].

So far, the clonal trees from the Scandes have not been genetically studied and compared to the nearby contemporary forest that grows at lower altitude and does not show unusual patterns of variation at the nuclear level[5,26,28]. Therefore, we conducted a genetic study of the clonal trees from the Scandes combined with ancient DNA studies from multiple lake and peat sites in northern Europe, to explore and better understand their origin and contribution to the recolonization of the postglacial landscape of Fennoscandia. Indeed, recent studies have shown that ancient DNA extracted from lake sediments (*sed*DNA) is a very powerful tool to detect locally established plants analogous to macrofossil remains[29,30]. S*ed*DNA is not derived from long-distance transport like pollen, which makes interpretations in respect to local occurrences straightforward. Pollen records, in contrast, carry local and regional signals, which can be difficult to disentangle.

We combined *sed*DNA data from 15 lake sediment and peat records from Fennoscandia with modern nuclear (~15,000 SNPs) and mitochondrial (mh05 locus) DNA data from clonal and forest spruce populations from the central Scandes, to scrutinize the genetic origin of the early colonizing trees of this region. Six of the *sed*DNA sites were selected from the central Scandes where previous findings of ancient wood were made and where the clonal trees are growing, two sites were in southern Sweden, four sites in Finland and two sites in Russia. We show that spruce was indeed present in the early Holocene in central Sweden and that a genetic similarity exists between the early colonizing clonal trees and the surrounding spruce forest. The most likely interpretation of these results is an early spread of trees from the east as land areas had become ice free, likely via seeds transported over

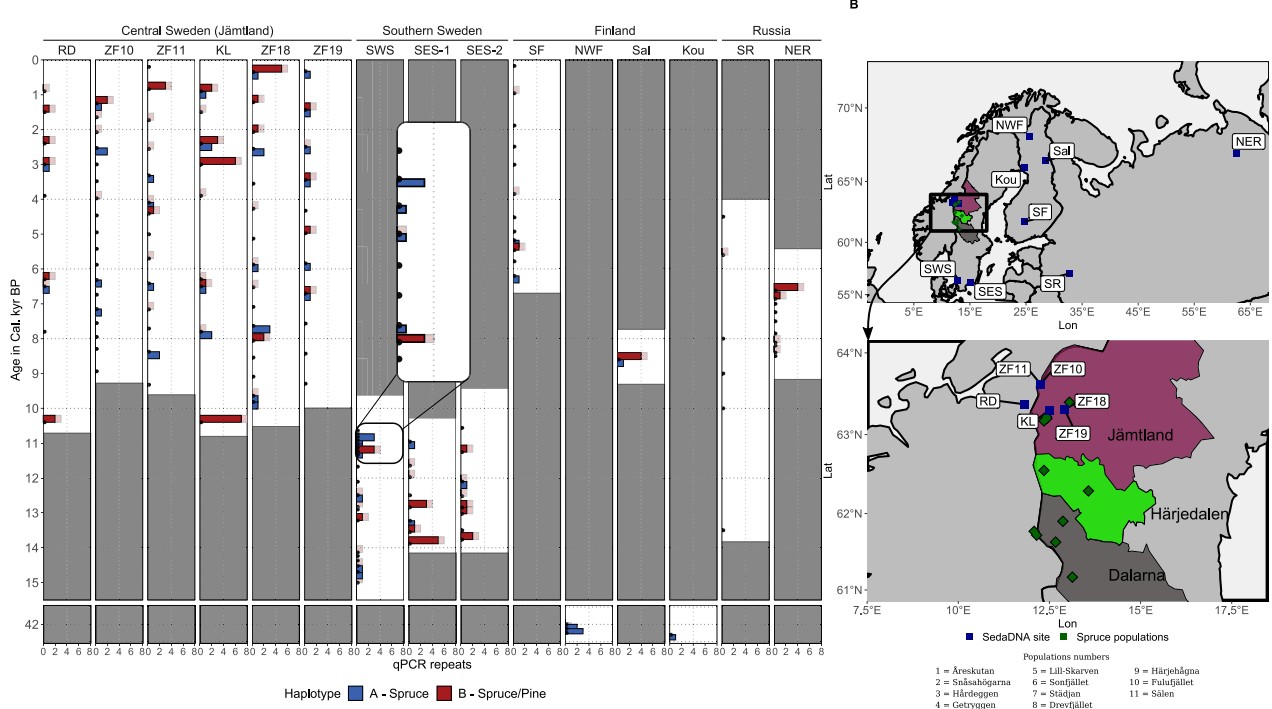

**Fig. 2 Summary of *sedDNA* results obtained using the mitochondrial mh05 marker. A** qPCR results on ancient sediments, with age on the y-axes and number of qPCR repeats on the x-axes. Mitochondrial mh05 haplotype A (blue) identifies only Norway spruce and haplotype B (red) identifies either Norway spruce or Scots pine. The transparent red bars represent haplotype B detections that are less certain due to background contamination. Black dots indicate analysed samples, and all the grey marked areas are time windows where no samples were analysed. **B** locations of all sites analysed: ancient *sedDNA* sites (blue) and modern spruce populations (green). Source data of panel A are provided as a Source Data file.

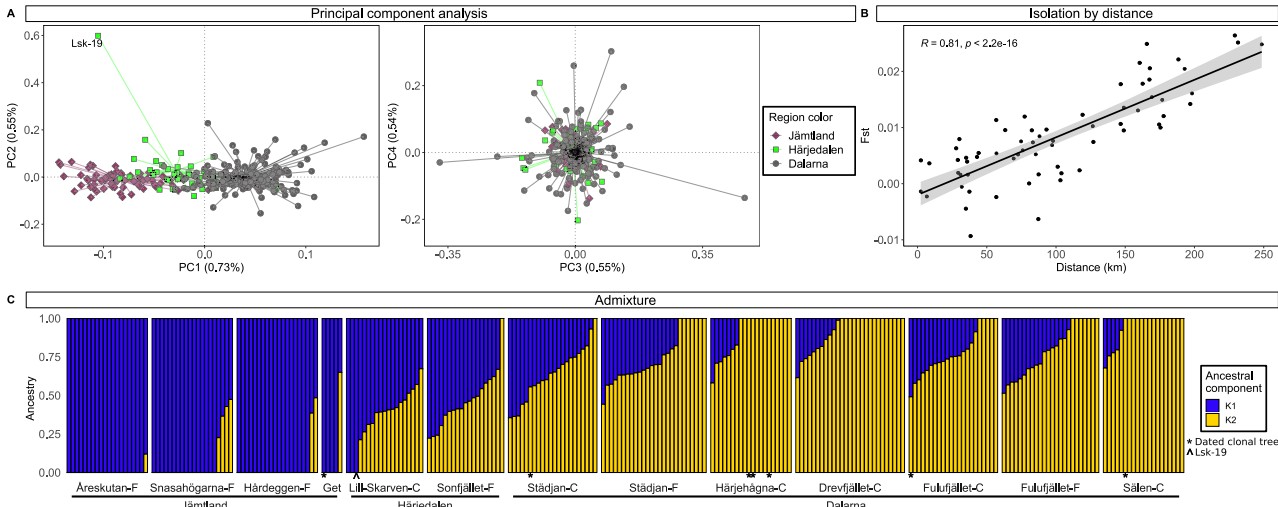

**Fig. 3 Overview of the results obtained using nuclear MIG-seq sequence data and 14,969 SNPs on contemporary spruce populations from the central Scandes. A** PCA analysis results (1st to 4th axes); PC1 shows a grouping based on geographical location with one outlier (Lsk-19) on the second axes. PCA axes 3 and 4 do not separate the populations further. In the PCA each point represents an individual coloured according to the sampling area. **B** Fst-values between populations plotted against geographical distances and showing a strong correlation (each dot in the plot represents a pairwise comparison between the populations given in Supplementary Table 2). Pearson correlation (two-sided) and Fitting Generalized Linear Model (GLM) was used to produce confidence interval. **C** admixture plots for K = 2 showing a declining proportion of the ancestral component dominant in the north (K1 = blue) in the more southern populations (K2 = yellow). Populations are ordered along latitude from the most northern population Åreskutan to the most southern population Sälen. Clonal trees are indicated with an asterisk and sample Lsk-19 with the ^ symbol. Source data are provided as a Source Data file.

kilometres on ice-crusted snow[25]. This was followed by a main east-to-west expansion during the late Holocene, which is well supported by previous pollen studies. Furthermore, by combining evidence from ancient and modern DNA we propose a third possible postglacial colonization scenario of Fennoscandia, as

argued below, with an early colonization event starting from a few spruce trees that persisted, likely at the southern and eastern margins of the SIS.

Here, we show that ancient *sed*DNA offers an important tool to reconstruct ancient environments following deglaciation, which

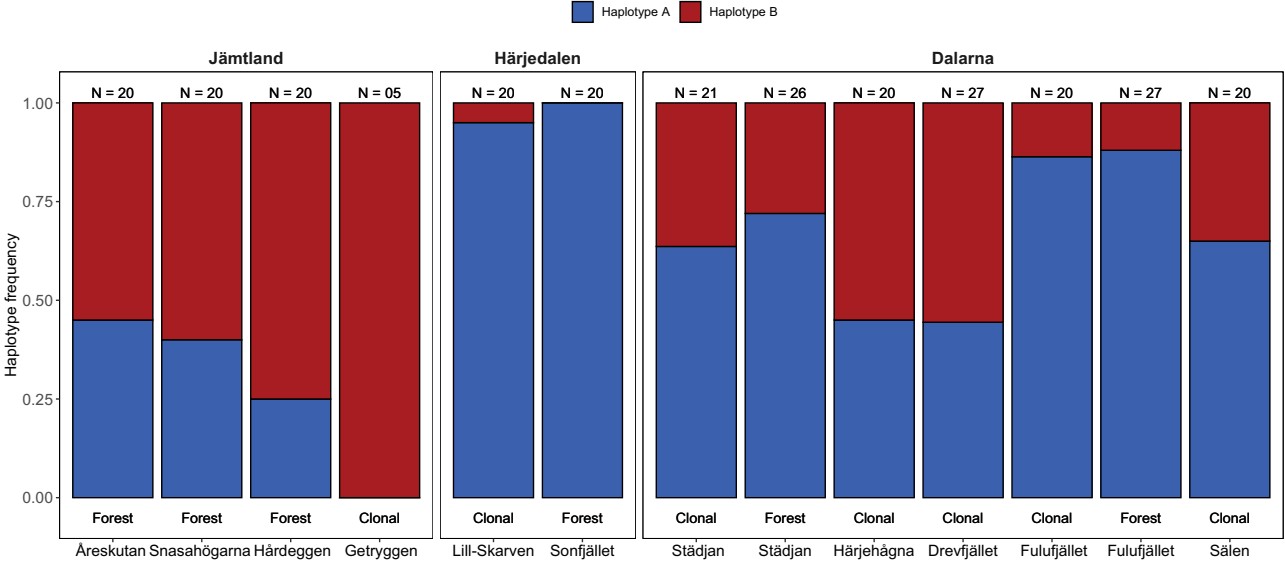

**Fig. 4 Mitochondrial mh05 haplotype frequencies (y-axis, proportion of individuals with either haplotype A or B within a population) in clonal and forest spruce populations ordered by latitude (x-axis).** Haplotype A (blue) is less abundant in populations from Jämtland County in the north and almost fixed in populations further south in Härjedalen County. The most southern populations from Dalarna County show a generally higher frequency of haplotype A. Clonal and surrounding forest populations shows almost identical haplotype frequencies. Population Getryggen consists of only five trees, the rest of the populations included between 20 and 27 trees as indicated for each population above the bars ($N =$). For each bar it is indicated whether a population was clonal or forest. Source data are provided in the Source Data file.

has important palaeoecological implications for the lateglacial environment in southern Sweden, particularly in relation to the ability of spruce to survive under changing climate.

## Results

**Ancient *sed*DNA Lateglacial and Holocene presence of spruce.** The mitochondrial mh05 haplotype A has so far been found only in *P. abies*, and it is therefore species-specific, while haplotype B is observed in both *P. abies* and *Pinus sylvestris* (Scots pine)[2,31]. Both haplotypes are absent in *Juniperus communis*, a third conifer native to central Fennoscandia. A fourth conifer has also been reported in the central Scandes between 8.7 and 7.5 cal. kyr BP (*Larix sibirica*)[32,33] but disappeared from the paleo record after 7.5 cal. kyr BP and is not present in Fennoscandia today. It seems unlikely that *L. sibirica* carries mh05 haplotype A because this variant shows a rare mutation with a deletion of 20 base pairs compared the common haplotype B detected in most of the sister species to spruce[2]. Therefore, we interpreted the detection of haplotype A in lake sediments and peat records as local presence of *P. abies*.

The overall detection of haplotype A from our 15 locations was low, with 54 positive qPCR reactions across samples, while haplotype B was more common with 72 detections (Supplementary Note 1). Out of the 1176 qPCR run on *sed*DNA samples, 54 showed the haplotype A (Fig. 2A) and this haplotype was significantly less abundant in the control negative samples (one in 208 extraction negatives, and one in 272 qPCR negatives, respectively). Haplotype A was found throughout the Holocene in all six lakes from the central Scandes (Fig. 2A), except in the two oldest samples (10.3 cal. kyr BP) ($n = 16$ qPCR) where we recovered only haplotype B. Outside Sweden, haplotype A was detected in samples from southern Finland (two samples dated ~5–7 cal, kyr BP) and in east Finnish Lapland (one sample dated 9.3 cal. ky BP), but was absent in Russia (SR and NER) (Fig. 2A, B). Both haplotypes were present in Lateglacial samples (~15–12 cal. kyr BP) from southwest and southeast Sweden (SWS, SES). The two oldest sediment sampling locations from

northwest Finland (NWF) and east Finnish Lapland (EFL-Kou, 43 cal. kyr BP) also showed haplotype A (Fig. 2A).

**Modern DNA clonal and forest trees are genetically close.** We obtained nuclear single nucleotide polymorphism (SNP) data (MIG-seq)[34] spread across the genome of 264 contemporary spruce trees from the central Scandes (Fig. 3) (135 clonal and 129 forest trees). After SNP calling and filtering, we retained 23,130 SNPs, of which 14,969 were unlinked ($r^2 < 0.2$). Population structure was consistent with the location of samples along a latitudinal transect where two ancestral clusters could be discriminated (Fig. 3A and C). Samples from the north (Jämtland County) contain a high proportion of the northern cluster, with trees in the south having more ancestral components from the southern cluster and generally less admixture in the most southern populations (Dalarna County). These results were in accordance with previous genetic findings showing two separated and widespread spruce clusters with a contact zone overlapping with our sampling area[27]. As expected, given the clustering, we also found a strong pattern of isolation by distance, with pairwise $F_{ST}$ values strongly correlating with geographic distances (Pearson's $r > 0.8$, Fig. 3B). Most importantly, neither the PCA nor the admixture analysis discriminated the clonal from the surrounding forest trees and the pattern was robust despite missing data and an uneven number of samples per cluster (see Supplementary Note 2, Supplementary Tables 4, 5, and Supplementary Figs. 3–7). Only one clonal tree in the Lill-Skarven population (Lsk-19) grouped away from the main cluster on the second principal component (Fig. 3A).

The mitochondrial mh05 shows a similar pattern as the nuclear SNPs data (Fig. 4) with haplotype frequencies for the two variants identical between clonal and surrounding forest populations. However, the frequency of mitochondrial haplotype A did not follow a clear latitudinal gradient as observed with nuclear SNPs. We found that haplotype A was nearly fixed in the two central populations from Härjedalen (Fig. 4), while it was present at high frequency (0.88) in the southern Fulufjället population and at

lower frequency in the populations from Dalarna (0.44–0.65, Fig. 4). The lowest frequencies (0.25 and 0.45) were found in two of the three northern populations in Jämtland. The population at Åreskutan showed a haplotype frequency like the Dalarna populations from Härjehågna and Drevfjället. Regarding the individual clonal trees that were dated, four out of seven trees showed haplotype A: Old Tjikko (9.5 cal. kyr BP, Fulufjället), GS Spruce (6.3 cal. kyr BP, Härjehågna), an unnamed tree (4.8 cal. kyr BP, Härjehågna) and Linnégranen (4.4 cal. kyr BP, Städjan). The remaining dated clonal trees showed haplotype B: Old Pompe (6.4 cal. kyr BP, Getryggen), Lindqvists gran (7.9 cal. kyr BP, Sälen) and an unnamed tree (7.9 cal. kyr BP, Härjehågna).

## Discussion

Our ancient sedDNA data suggest that *P. abies* was present throughout the Holocene in central Fennoscandia and during the Lateglacial in southern Sweden. These findings are in line with the oldest (13 cal. kyr BP) recovered spruce macrofossil found in the Scandes[13–15] and supports a scenario where spruce was present in southern Sweden as early as 14.7 ± 0.1 cal. kyr BP and established in the central Scandes around 10 cal. kyr BP. In the central Scandes, spruce trees remained present throughout the Holocene, likely producing little or no pollen but leaving detectable DNA in the sediment. This early presence of spruce in southern Sweden has been questioned because of low levels of spruce pollen identified in pollen stratigraphic of older studies[21]. While our data support the early finds of spruce pollen, we cannot draw conclusions regarding the continuous presence of spruce in southern Sweden due to a gap in our chronology (grey zones in Fig. 2). Nevertheless, we retrieved positive amplifications for spruce in all three sediment cores from central-southern and we can securely exclude contamination as a cause of our finding, as it is very unlikely that a rare haplotype (haplotype A) would contaminate material during subsampling or analysis (see Supplementary Note 3 for details). In addition, samples from the lakes were collected during different expeditions, and core opening occurred at different locations in areas where haplotype A was totally absent or very rare. It is also very unlikely that contamination occurred following a specific pattern (i.e., present at multiple sites but not in those distant from the Scandes) (see also Supplementary Note 3 and Supplementary Table 6 for data simulations to exclude contamination).

DNA and pollen from the Marine Isotope Stage 3 interstadial (MIS 3, 60-27 kyr BP) can be preserved in the upper few decimetres of glacial deposits underlying lake sediments[35]. Therefore, samples located at the interface between the lake and underlying glacial sediments need to be interpreted with caution as they may contain reworked material. Accordingly, we did not analyse or interpret DNA located in basal lake sediments, as a precaution to avoid reworked material.

The presence of spruce DNA at 14.7 ± 0.1 cal. kyr BP in south Swedish lake sediments pushes the earliest record of spruce in the lateglacial landscape of Fennoscandia back by ~1.7 kyr BP. Two of our sites in southern Sweden (SWS and SES) were previously investigated using multiple palaeoecological proxies[36–39], including a published ancient DNA record from one of the lakes (SES)[40], but none of these studies suggested the presence of spruce. The ancient DNA record, however, had almost no chance of recovering spruce among the billions of reads sequenced from sedDNA because it used the shotgun sequencing approach without capture, i.e., a method with extremely low coverage for plant reads and consequently a low possibility of sequencing rare species. The fact that spruce pollen or macrofossils were not recovered at these sites is not surprising, because a small, sparse population of trees, likely using clonal reproduction strategies, do

not produce enough pollen to be detected and are less likely to leave large quantities of macrofossil remains. Moreover, the limited sediment material available for these studies did not provide sufficient material for extensive macrofossil analysis. Ancient sedDNA, is of local origin[30], it binds to and is protected by fine-grained mineral particles in sediments and provides a proxy for plant paleo-community reconstruction that exhibits more similarity to macrofossils than to pollen, and thus offers the opportunity of detecting 'cryptic species' that do not produce pollen.

The early presence of spruce in southern Sweden is supported by the pioneer work of Nilsson in 1935 who found spruce pollen in 11 Oldest Dryas (about 16-17 cal. kyr BP) peat deposits in southern Sweden[41,42]. However, the limited number of total grains found in many of these studies has been used as an argument to dismiss the findings, especially since successive sedimentary records were showing relative abundances of spruce pollen below the typical threshold assumed to indicate local spruce trees[21]. Moreover, these early pollen records[41] predate accurate carbon-14 dating. Nevertheless, the high frequency of spruce pollen detected by Nilsson (up to 11% of all tree pollen) made him suggest that spruce was locally present during the Lateglacial in southern Sweden. Moreover, previous studies show a high July summer temperatures and continental climate throughout the Lateglacial (15–11 cal. kyr BP) in southern Sweden[39,43], and at the specific SWS site the lake catchment stabilized between 14.7 and 14.5 cal. kyr BP[38]. Stable soils and warm summer temperatures could have been favorable for spruce to colonize the area, even though spruce is not a species particularly environmental demanding, young trees can be prone to spring frost[44]. In arctic Russia spruce is the treeline species, but it grows equally well in small pockets in the tundra as well on permafrost.

Lateglacial (13–11 cal. kyr BP) findings of spruce macrofossils in the Scandes generated the hypothesis of an early colonisation of Fennoscandia from the west[13]. The theory was later reinforced by Early Holocene (10.3 kyr BP) findings of spruce (haplotype A) in sediments from Trøndelag in central Norway (~100 km from our Jämtland sites in Sweden)[2]. In our study, we found haplotype A in multiple qPCR repeats from lake sediment samples in central Sweden dated ~10 cal. kyr BP and in sediments deposited at two independent locations prior to the Last Glacial Maximum (LGM) at ~43 cal. kyr BP in northeast Finland. These findings imply that haplotype A is not restricted to a western refugial population as previously hypothesized[2], as it is also present east of the SIS, at least during parts of the MIS 3 which was a long period including warm and cold intervals[45]. Interestingly, our sedDNA findings also suggest that this haplotype was present in Finland (two sites) during the early and mid-Holocene but disappeared from our younger sedimentary records. The absence of haplotype A in younger sediments and in contemporary forests of north-east Finland, together with its early detection in the central Scandes, suggest that it disappeared from the north-east and followed the retreating ice sheet into central Sweden.

Even though macrofossil of spruce from the MIS3 interstadial material in northeast Finland are needed for more in-depth knowledge regarding early spruce in this region, based on our results we suggest that spruce may have been present during parts of the MIS 3 close to the ice sheet margins, possibly northwest Russia. Our DNA finds are only indirectly supported by the fact that bioclimatic conditions might have been suitable for spruce around this time. The latter is evident considering multiproxy data and macrofossil findings from Finnish Lapland, indicating average July temperatures in some periods were as high as 14.4 °C and the presence of a birch-pine forest during periods in the MIS3 interstadial[46]. Bioclimatic model approaches even suggest

suitable spruce habitats outside the eastern SIS margin during the colder LGM period[47].

The ages of our spruce clones (4–9.5 cal. kyr BP) in the central Scandes make them living remnants of the spruce populations that colonized Fennoscandia before the late Holocene east-to-west expansion wave (arriving to the Scandes ca. 2–3 cal. kyr BP). Because we found haplotype A throughout the Holocene in lake sediments at higher elevation in our study region, we are confident that there is a genetic continuity either directly, or indirectly due to sexual reproduction, between the old fossil remains found below these old trees and the still-living trees we sampled for this study. Given the long-term cooling trend and descending treelines observed during the last 3000 years[12], it is very unlikely that these trees—currently still located above the treeline—arrived during the late Holocene and dispersed from lower altitudes (see extra discussion in Supplementary Note 4). Our sampled clonal spruce populations show a higher level of genetic similarity with the closest surrounding forest than with the clonal populations at further locations, with a clear pattern of isolation by distance in line with results from a previous genetic study conducted in the area[27]. Because spruce is a slow evolving species with an overall low level of genetic diversity[48], the short time interval passed between its early and late arrival in Fennoscandia (ca 8 kyr) is not long enough for mutations to occur and accumulate. Hence, it seems likely that the early established spruce clones share the same genetic origin as the contemporary local forest, which originated from glacial refugia in the east[5] [we found only one genetic outlier (Lsk-19, Fig. 3A) and this separation disappears with more stringent filtering (see Supplementary Note 2 and Supplementary Fig. 3). Indeed, if the clonal trees had a common source and established prior to the arrival of the late Holocene migration from the east, they would share more resemblance to each other than to the surrounding forest. Our interpretation is therefore that the ancestors of the studied clonal trees from Dalarna (the yellow ancestral component K2 in Fig. 3C) migrated from populations located in south-eastern Finland, which arrived earlier from Russia. Potentially trees also arrived from southern Sweden. Trees in Dalarna may have become established as early as ~9.5 cal. kyr BP. The clonal trees from Jämtland instead belong to a second genetic cluster (the blue ancestral component K1 in Fig. 3C), which also originated from the east, but their ancestors may have migrated through northern Finland.

While we do not argue for a LGM tree refugia south of the Scandinavian ice sheet, we speculate that the early presence of spruce in southern Sweden is best explained by a local survival of trees close to the ice sheet margin rather than an early immigration from the Baltic[25]. Although, macrofossil finds close to the Baltics, from Belarus (south of the SIS) indicate spruce to be present[49]. The latter scenario would ignore the underling genetic variation existing today in the Baltic populations. Indeed, modern spruce populations from the Baltic region do not show the short mitochondrial haplotype A[2] and based on nuclear DNA data[5] they cluster separately from the Scandinavian populations, and there is also no evidence that the Baltic populations were replaced after they had become established. We acknowledge that the hypothesis that spruce trees survived at the southern margin of the SIS close to southern Sweden (Denmark and/or Doggerland where few spruce pollen grains were found from the Younger Dryas[50]) lack supporting macro and mega-fossil evidence. More studies are required to investigate the early presence of spruce in this region, as this will have implications also for other plant species.

Alsos et al.[51] recently used sedDNA to test the hypothesis of an LGM refugium in Andøya (northern Norway) and although results were not conclusive for proving the presence of spruce in the region, the authors concluded that periods existed when the climate in Andøya was warm enough for Norway spruce and Scots pine to grow. Similarly, our results, cannot exclude that some trees from the central Scandes originated from individuals surviving west of the SIS, such as Andøya, or the Trøndelag region[2]. However, our modern DNA analysis suggest that if they survived in the west, their genetic impact on the recolonization of Fennoscandia was negligible because we observed no deviation in these trees compared to the current and widespread, ancestral genetic clusters[28].

Based on our DNA results we picture the establishment of Norway spruce in central Fennoscandia during the Lateglacial to Early Holocene transition. The genetic signature we found in the sampled clonal and forest populations suggest that spruce trees in this region belong to two wide-spread genetic clusters present in northern Europe[28]. We suggest that postglacial recolonization of Norway spruce in Fennoscandia started already during the Lateglacial and that mechanism(s) for seed dispersal at this early time remain unknown. Possible mechanisms may have involved long-distance seed dispersal events from eastern populations by water transport across the Baltic Ice Lake, e.g., a dispersal mechanism that drives vascular plants colonization of contemporary remote arctic regions[52]. Seeds transported from Finland on ice-crusted snow and ice seems an additional possible driver of early spruce dispersal. Early spruce colonizers in the central Scandes maintained themselves in populations of likely small sizes and expanded from east to west only during the Late Holocene. The proposed migration scenario does not require a distinct western glacial refugia[2].

## Methods

**Ancient material and sedDNA extraction.** We used sediment and peat records from 13 locations within Fennoscandia and two locations in Russia (Fig. 2B, Supplementary Table 1). Nine of the sites covered the Holocene period (five in Sweden, one in Norway, one in southern Finland and two in Russia). Two sites in southern Sweden covered the Late-glacial and Early Holocene periods (15.3–9.5 kyr BP) and one site from northwest Finland dates to 42 cal. kyr BP. Palaeoecological and site descriptions on all the sites have been published elsewhere (see Supplementary Table 1 also for dating and references). The sites in southern Sweden, Finland, and Russia were selected because the material was already available from previous studies and because interesting for investigating spruce recolonization.

All sediments were subsampled in clean rooms and the most sheltered internal part of the sediment was used for DNA extraction to reduce the risk of contamination. For each sample, approximately 0.25-0.35 g of material was homogenised by vortexing before DNA was extracted using the DNeasy PowerSoil extraction kit (Qiagen, Hilden, Germany) with the following modifications: bead-beating was done by vortexing tubes horizontally at the highest speed (Fisherbrand, 3000 rpm) for 10 min and 2 μL proteinase K (20 mg/mL) and 25 μL DTT (1 M) were added to the PowerBead tubes after bead-beating[53]. The PowerBead Tubes were incubated at 37 °C, rotating overnight. Solution C3 was increased to 250 μL and up to 800 μL of the supernatant was added to 1400 μL on solution C4. DNA was eluted twice in 65 μL elution buffer containing 10 mM Tris-HCL and 0.05% Tween-20 and incubated for 10 min at room temperature (collection in the same tube). All DNA extractions were done in a laboratory specifically dedicated to aDNA analyses at Uppsala University following established procedures[29]. Extractions were done in batches of maximal 20 samples and contained two extraction blanks (extraction without sediment/peat). Extraction blanks were processed in the same way as the samples.

All the sedDNA extracts were checked for inhibition by spiking qPCR reactions with a synthetic control oligonucleotide and then amplified with primers specific for this template[54]. DNA extracts were diluted until the amplification curve was nearly identical to the positive control. The qPCR reactions contained the following: 0.5 μM SynFor and SynRev primers, 10e-9 μM control template, 1x TATAA SYBR GrandMaster Mix, and 1 μL DNA extract (diluted until expected amplification) in 10 μL reactions. The qPCR thermocycling protocol was as follows: 95 °C for 30 s, 50 cycles of 95 °C for 5 sec, 55 °C for 30 sec, 72 °C for 10 sec. A melting curve was created starting from 60 °C to 95 °C with 0.5 °C increments. This test only gives an indication whether the sample is diluted enough to remove the effect of inhibitors because different PCR template and primers might be differently affected by inhibitors than the control template.

**Modern spruce material and DNA extraction.** We collected needles from seven *P. abies* populations, which in three cases included also unrelated groups of clonal trees. We considered a clone one individual with one or several stems regenerated

from a common root system not more than 2–3 m tall. These aboveground parts will eventually blowdown or die from other causes, but the clone is still alive. All clonal trees were located at or above the current treeline of the continuous pine (*Pinus sylvestris*) and spruce-dominated forest (between 765 and 1004 a.s.l.) in the central Scandes. Six populations were in Sweden (Jämtland and Dalarna Counties) and one in Norway. In each population, we sampled needles from at least 19 different clones, except for the Getryggen population where we only found five clones. All sampled clones consisted of individual stands in alpine heath vegetation and sometimes more wet vegetation at the margins to alpine mires. Most trees had a rich abundance of epiphytic lichens and looked healthy. The trunk ages of the clonal trees have been reported to range from ~100 to 600 tree rings[14,55]. For three of the seven populations (Fulufjället, Städjan, and Getryggen), the surrounding nonclonal spruce forest at lower elevations (below the current treeline) was also sampled. Other nonclonal populations were also sampled (Snåsahögarna, Åre-skutan, Hårdeggen and Sonfjället). In total, we sampled 113 clonal trees above the treeline and 132 nonclonal trees from 13 populations. Seven of the 113 sampled clonal trees had been previously radiocarbon dated with ages ranging from 4.4 to 9.5 cal. kyr BP (Supplementary Table 2). The ages are established based on the radiocarbon dating of macrofossils found directly under the still living clonal trees. None of the sampled modern spruce trees showed unexpected levels of SNP variation suggesting they were planted from different provenances.

We used two *Picea obovata* samples and eight *P. abies* samples from four well discriminated genetic clusters (Carpathian, Russian-Baltic, Central Europe, and Alpine[5]) as external reference to help place the Fennoscandian samples in a larger genetic context.

DNA was extracted from needles from all clonal and non-clonal trees using the DNeasy Plant Mini kit (Qiagen) following the manufactures recommendations. Before starting the extraction, five young needles were mechanically disrupted as follows needles were frozen in liquid nitrogen together with two 2 mm sterile metal beads for 30 s and ground using the Tissue-Lyser II set on 21 Hz for 30 sec (Retsch mixer mill MM 300), this was repeated twice.

**Mitochondrial DNA analysis on ancient material**. For genotyping the mito-chondrial mh05 fragment, a qPCR melting curve assay was developed to accurately distinguish haplotype A and B characterised by a 21 bp deletion (see Supplementary Note 1, Supplementary Tables 7–8 and Supplementary Figs. 1-2 for validation of the method). Because we were working with aDNA that is always fragmented and damaged[56], primers were redesigned to make the two amplicons as short as possible (64 bp and 85 bp, respectively). Recovering haplotype A, indicates the presence of *P. abies* because this haplotype has not been found in the two other conifers native to central Fennoscandia: Scots pine (*Pinus sylvestris*)[31] and common juniper (*Juniperus communis*. For genotyping the mh05 fragment, the qPCR reactions contained the following: 0.5 µL mh05shortFOR and mh05shortREV primers, 1x TATAA SYBR GrandMaster Mix, and 1 µL DNA extract (diluted according to the inhibition test, see Supplementary Note 5 and Supplementary Fig. 8) in a total volume of 10 µL. Each qPCR was repeated eight times[57] and each qPCR run contained eight PCR template negatives to monitor contamination. The qPCR thermocycling protocol was as follows: 95 °C for 30 s, 60 cycles of 95 °C for 5 s, 55 °C for 30 s, and 72 °C for 10 s. A melting curve was created starting from 60 °C to 95 °C with 0.2 °C increments. Haplotype A or B was recorded as present if the melting peak showed a clear melting curve at 74 ± 0.6 °C or 76 ± 0.6 °C respectively. A subset of the qPCR reactions that yield haplotype A and B were sent for Sanger sequencing to confirm sequence identity (Supplementary Note 6 for Sanger sequencing preparations). The average number of positive reactions found in the negative controls was subtracted from the samples.

**Nuclear DNA analysis on modern material**. Genome-wide SNP genotyping on clonal and forest spruce trees was done using the PCR-based method for genome-wide single-nucleotide polymorphism (MIG-seq) as described by Suyama and Matsuki[34]. In short, this method uses eight primer pairs that target inter-simple sequences (ISSR) and is used to PCR amplify de novo genomic regions, which can bioinformatically screened for SNPs. MIG-seq library was prepared via a two-step PCR with one minor modification from Suyama and Matsuki[34] (the annealing temperature of the first PCR was lowered from 48 °C to 38 °C). In the first PCR, Inter-Simple Sequence Repeat regions were amplified using MIG-seq primer set 1. Then, the second PCR was performed to add the adapter and indices for Illumina sequencing. Sequencing was done on an Illumina MiSeq using 150 cycle paired-end run.

The raw reads (19,095,149, mean per sample 68,936 ± 12,900) were mapped to the spruce genome reference v1.0[58] using the default settings of bwa-mem[59]. All mapped positions with a coverage below half of the number of samples were removed. SNP calling was performed with GATK v4.1.4.1. Raw SNPs were filtered as follows by excluding SNPs with: AD (alternative allele count) <3, MQ (RMS mapping quality) < 45, QD (variant confidence/quality by depth) < 0.8, QUAL < 20 SOR (symmetric odds ratio) >3, AF < 0.03 and AF > 0.97 (Alternative allele frequency). Loci in too high linkage disequilibrium $r^2 > 0.2$ were removed using PLINK2[60] and loci assigned to the chloroplast genome were removed.

To control for potential informative missingness/absence in the data, downstream analyses were repeated using all SNPs or excluding those with more than 25% and 50% missing data. In a first step, the *SmartPCA* function from Eigensoft[61] was used to perform a principal component analysis (PCA) using all samples.

For further analyses, the non-Fennoscandian samples ($n = 10$) were removed from the dataset because they could mask weak population structure within the Scandian populations. Small-scale population structure was characterized using the *SmartPCA* and the individual ancestry component was computed using Admixture[62]. Pairwise $F_{st}$[63] between each pair of populations and the associated *p* values (1000 bootstraps) were calculated using the StAMPP package[64] and geographical distances between the populations were calculated using the *distm* function from the package geosphere[65] in $R$[66]. All plots were created with ggplot2[67].

**Mitochondrial DNA analysis on modern material**. The same qPCR melting curve assay as used for genotyping the ancient samples was used for the modern spruce samples. The qPCR reactions contained the following 0.5 µL *mh05shortFOR* and *mh05shortREV* primers, 1x TATAA SYBR GrandMaster Mix, and 1 µL of 1:100 diluted DNA extract in a total volume of 10 µL. The qPCR thermocycling protocol was as follows: 95 °C for 30 s, 40 cycles of 95 °C for 5 s, 55 °C for 30 s, 72 °C for 10 s. A melting curve was created starting from 60 °C to 95 °C with 0.2 °C increments. A sub-section of the samples that showed inconclusive melting peaks and samples showing double peaks were Sanger sequenced using primers amplifying a larger part of the mh05 locus[2] (see Supplementary Note 1; Supplementary Table 3).

**Reporting summary**. Further information on research design is available in the Nature Research Reporting Summary linked to this article.

## Data availability
The raw qPCR melting curve data and the summarised data used to create Figs. 2a and 4 are available in Source Data file. Additional raw qPCR data and Sanger sequences that confirm sequence identity are available in figshare [https://figshare.com/articles/dataset/Raw_qPCR_data_and_raw_Sanger_sequences_/14837859]. Trimmed Sanger sequences are also available in Source Data file. The distinctive character of haplotype A and B obtained by Sanger sequences are identical to accession HE652907.1 and HE652882.1, respectively. The raw high throughput sequencing data, underlying source data for Fig. 3a-c, are available in DDBJ DRA with the identifier DRA012297. The *Picea abies* v1.0 refence genome[57] used from mapping is available at plantgenie.org [https://plantgenie.org/FTP?dir=Data%2FConGenIE%2FPicea_abies%2Fv1.0%2FFASTA%2FGenomeAssemblies]. Source data are provided with this paper.

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

## Acknowledgements

We thank Anne van Woerkom (Funded by the Swedish research council 2017-04548), Mattias Tassel and Elisa Pierfederici for assisting the molecular work; Sofia Ninnes and Eric Capo for fieldwork, subsampling, and radiocarbon dating of the Jämtland lakes with support from the Knut and Alice Wallenberg Foundation; Pertti Sarala and for providing samples used for ancient DNA analysis from north-western Finland. The computations and data handling were enabled by resources provided by the Swedish National Infra-structure for Computing (SNIC) at Uppmax (Uppsala) and partially funded by the Swedish Research Council (nr. 2018-05973). Fieldwork and molecular work for and modern and ancient DNA was financed with grants obtained from the Swedish Phyto-geographical Society and the Extensus Foundation.

## Author contributions

L.P. and K.N. conceived the study. L.P., K.N., A.N., and U.G. collected the spruce needles. K.N., A.N, T.S., and A.M performed the molecular analyses. J.K., R.B., H.S., M.M.V., and B.W. provided the ancient sediment material. K.N., P.M., and H.S.K. performed the SNP analysis. Y.S. organized and financed the sequencing. K.N. and L.P. interpreted the data

with contributions from J.K., P.M., R.B., B.S., S.B., H.S., M.M.V., and B.W. K.N. and L.P. wrote the manuscript with significant contribution from J.K. All authors discussed the results and commented and edited the last versions of the manuscript.

## Funding

## Competing interests
The authors declare no competing interests.
