## [Peer Review File · Nature Communications]

Reviewers' Comments:

Reviewer #1:

Remarks to the Author:

The manuscript provides a wealth of new data which is all interesting. While some finds can be discussed controversially others straight away help gaining a better understanding of the past. It would do the data justice to publish it in separate publications to be able to discuss the finds and implications. In the current format it is difficult to see how the data support the interpretations. The data from extant populations add strongly to the understanding of the history of *Picea* the aDNA data needs further discussion and interpretation. Both are not resolving the debate as promised in the title.

Please see the uploaded PDF for detailed comments.

Comments on the manuscript “Resolving Norway spruce origins during the glacial to Holocene transition in Fennoscandia”

Preface

The postglacial history of spruce in Scandinavia is a much researched and discussed topic that seems difficult to solve. In addition to my review of the fossil evidence (Giesecke and Bennett 2004 [doi.org/10.1111/j.1365-2699.2004.01095.x]), the publications by Tollefsrud et al. [Molecular Ecology (2008) 17, 4134–4150; Heredity (2009) 102, 549–562] have yielded insights on the genetic structure of extant populations that elucidate the colonization of *Picea*. Based on this and a fresh look at the fossil evidence I have compiled the below maps when discussing the topic after the publication by Parducci et al. 2012 in Science. I included the maps in a presentation at the IPC/IOPC in Tokyo, Japan in 2012 and a few conference talks thereafter. Although, these maps were never published in a scientific paper, underlying ideas are included in Birks, et al. (2012) [Science doi.org/10.1126/science.1225345] Giesecke, T. (2013). [The Encyclopedia of Quaternary Science, vol 3 pp. 854-860 Elsevier] and Giesecke et al. 2017 [Journal of Biogeography doi/pdfdirect/10.1111/jbi.12974]. I am sure that the senior author and several co-authors are aware of this and find it a pity that these interpretation finds no mention in the manuscript.

Tentative Lateglacial and early Holocene distribution of Picea at low abundance with each tree making a site with presumed presence of the tree in the area and smaller symbols indicate high uncertainty.

General comments on the manuscript

I very much welcome the conducted analysis and appreciate the new and interesting results presented here. Personally, I would have liked to see the material divided into two more than one manuscript presented for more specialized journals. Due to the different investigations on fossil and extant material, I find the argumentation sometimes difficult to follow. Given the long history of the debate and the involvement of the senior author I find it a pity that some literature is wrongly cited or omitted (see below).

I have two general problems with the interpretation of the results:

1) No evidence of a survival south of the ice: When compiling the above maps, I searched for the smallest amount of pollen evidence in Lateglacial sites south of the ice and found none. The here presented interpretation that *Picea* may have survived the LGM south of the ice seems to be based

solely on the results of a species distribution model based on one or two climate model results (Svenning et al. 2008 [Journal of Ecology 96(6) doi.org/10.1111/j.1365-2745.2008.01422.x]), which are unlikely possibilities lacking any supportive evidence.

2) Age of oldest *Picea* detection in S-Sweden: My other main concern is the age of the *Picea* detection on the Kullen Peninsula, which at 15000 cal. BP reminds of the situation of Andøya. The first Birch macrofossils detected at that site date to about 14500 cal. BP (after the Bölling warming), which is also very early considering that the earliest finds in Denmark date to about 13700 cal. BP (Mortensen et al. 2014 [DOI 10.1007/s00334-014-0433-7]). Pushing the earliest occurrence of *Picea* in southern Sweden to before the Bölling warming and into a landscape completely bare of other trees is a significant finding that deserves a critical discussion as it has implications. The Kullen Peninsula was covered by ice during the LGM so here we are not talking about a survival of *Picea* but the spread of the tree to that place after deglaciation and before the Bölling warming. With regards to the *Picea* detection in the core from Hässeldala, I am also surprised that an earlier publication on ancient DNA from a core of that site did not detect any *Picea* or *Pinus* DNA for the Lateglacial or Early Holocene (Parducci et al. 2019 doi.org/10.3389/fevo.2019.00189). At this site earliest tree birch evidence seems to be dating to 13300 cal. BP, which is more in line with known regional vegetation history. Also here *Picea* would have established before *Betula* or other boreal trees, which needs a discussion.

Specific comments

L. 44-47: The pollen diagrams reviewed in the studies 5,6 are much of the same as used in studies 7,8. There are few new (last 20 years) palaeoecological studies available from S-Sweden and no overview using the newer data. Study 8 is not showing early *Picea* pollen in S-Sweden.

L. 53: Why are you here not using the oldest published date of Kullman (2002), which you do cite?

L. 55: Based on your earlier age in this sentence the time gap would be much shorter than 10,000 years. Also this is a very interesting question (see also Giesecke 2004 <https://www.diva-portal.org/smash/get/diva2:165274/FULLTEXT01.pdf>) but you are not addressing it later.

L. 58: Please be more specific of what you want to discuss here as an “establishment prior to deglaciation” is problematic as the tree is not growing on ice.

L. 99: You don't provide evidence for a survival near the southern and eastern margin of the ice. I much question the survival in the south if the ice.

L. 118: See general comment. These findings need a more extensive discussion.

L. 151: The cited publications claim older ages for the presence of *Picea* in the mountains.

L. 154: Which regions do you refer to here – central and southern Sweden? If so what is the evidence for continued occurrence of *Picea* in southern Sweden?

L. 156: The reference “13” is questioning the survival of trees in Norway related to the finds by Kullman (2002) in the mountains.

L. 157-158: I am not disputing the presence of the tree during these times. Nevertheless, the fact that you so readily find it in most lakes is surprising as pollen evidence is very sparse in samples dating to the Early Holocene at most sites and if the tree was standing near a lake so that its DNA is preserved also the pollen should be there. I investigated 4 Sites in central Sweden scanning pollen samples for *Picea* pollen in Early Holocene samples and found hardly any (Giesecke 2005).

doi.org/10.1016/j.quascirev.2005.03.002). So were you simply luckier than I or can you explain this mismatch?

L. 169: Which “refugia east of the Weichselian ice sheet” do you refer to here?

L. 176-178: The geographical setting in North America is completely different so you should elaborate which parallels you see exactly.

L. 181: Yes, but trees were subsequently expelled from these locations so what is the relevance here.

L. 183: The study did not focus on *Picea* and the results regarding *Picea* are very questionably and not at all supported by any palaeoecological evidence.

L. 222-223: I don't see the logical connection of the argument here and I much dispute the survival from the south for which there is no evidence.

L. 227: I don't see what you resolve. The existence of the tree in Scandinavia during the early Holocene was already considered by Giesecke and Bennett (2004) and you don't present evidence for the survival near the margin of the LGM ice.

Thomas Giesecke

Reviewer #2:

Remarks to the Author:

Spruce is the most important tree species in Fennoscandia no matter what aspect: Economy, recreation, biodiversity, etc. Yet, its history is still shrouded in obscurity after more than 100 years of investigations and debate. The study is novel and confirms a very early presence of spruce in Scandinavia, which certainly is good enough for justifying publication in Nature Communication, and I recommend it to be accepted. However, it does not, as claimed, solve the question about spruce origin and migration routes and it needs a major revision before being accepted.

As I am not a geneticist, I cannot assess the quality of the genetic part in the study. I assume another reviewer with that expertise has been assigned. However, I know from discussions in the scientific community that there could be a large risk for contamination. The DNA is small, to say the least. I have no opinion in the matter, although with my novice perspective their arguments in the supplementary information seem convincing. Anyway, the text regarding this subject should not to be hidden there but to be discussed in detail in the main manuscript – and scrutinized by expertise in the field.

The very same wetland analysed here for ancient DNA (SWS) was also analysed by Wohlfarth et al (2018) for among other things pollen, macrofossils, hydrogen isotopes and chironomids. In contrary to Nota et al, their study does not at all support the presence of spruce at the time. Apart from finding no physical evidence of the tree, their reconstruction of vegetation and climate showed anything but a spruce-friendly environment. According to them, it was relatively warm 15,000 years ago, but there were no trees. Instead, it was an open arctic landscape dominated by herbs, grasses and dwarf shrubs. I would like the authors to comment on that and discuss possible reasons behind the (large) discrepancy between the studies.

Row 74-76 and 130-132. "Each if these clones...likely originating from a single ancient individual". Connected or not is crucial for this study. It has been questioned if the wood in the soil actually was connected to the living "Old Tjikko" tree. See <https://kimmerer.com/how-old-is-old-tjikko/> and references therein. If not, the same could be the case for all clonal trees in the study. I would like the authors to comment on the possible consequences for the study if they are not connected. If the living tree is of much younger origin than the wood in the soil, and sexually generated, it is not surprising the living trees share DNA with trees in the surrounding forest, and many of the conclusions in the paper must be revised.

It must be shown in diagram 2 which time periods that were sampled (e.g. remove the y-axis for the sections not sampled), as presented now you can get the impression spruce was absent although there is no evidence in either direction. As now, it is very difficult to interpret the evidences for spruce (haplotype A) presence or non-presence, and it is difficult to follow the authors discussion.

Row 192-onwards. What's the evidence that today's local forest originates from the east? I don't follow the logic here. As the clonal trees are more similar to the surrounding forests that came later than to other clonal trees in the mountains, it seems hard to believe there have been no contact between the new forest and the living (clonal) trees. How else can they be so similar, when they have been separated for thousands of years and thousands of kilometres? This raises the question whether the living trees (Old Tjikko for instance) actually are clones originating from the old wood in the soil. The study does not present convincing evidences or even indications, as claimed in rows 226-230, that spruce arrived to central Sweden early from the south and north. Neither does it explain the origin of the very early (up to 13,000 year old) spruce wood in the mountains.

Minor comments

Row 31. It is not north-east, but rather east.

Row 44-47. But there is one study with pollen that was considered to come from local origins: Segerström, U. and H. von Stedingk, Early-Holocene spruce, *Picea abies* (L.) Karst., in west central Sweden as revealed by pollen analysis. *Holocene*, 2003. 13(6): p. 897-906

Row 49. The oldest spruce wood in the Kullman publication is older than that: around 13,000 cal BP (Beta-121826 11,020 ±90 *Picea abies* Stem)

Row 54. "until" better than "only" according to me

Row 59. The directions here are problematic, does an eastern route mean it came east or migrated to the east? I assume it here means: "migrated to the west", or "a western migration route". In several cases in the text it is unclear whether the direction regards from where the spruce migrated or if it is in which direction it migrated to.

Row 61-62. Well, if it did come from the south, it left quite significant traces in the pollen record: Check table 1 in: Lindbladh, M., När granen kom till byn – några tankar kring granens invandring till södra Sverige (The immigration of spruce to southern Sweden). *Svensk Botanisk Tidskrift*, 2004. 98: p. 249-262.

Row 63-64. He first brought up the idea in 2002, in publication no 10 here.

Row 67. kyr, not yr.

Row 71. Citation mark missing.

Row 88-89. What if the DNA come from pollen? It is unlikely large pieces of wood end up on the lake bottom or on a mire, and pollen can make up pretty large quantities.

Row 92 and onwards. Somewhere here the choice of sites needs to be discussed. The mountain sites in Sweden are easy to motivate due to the findings of ancient wood. The other sites are perhaps less easily motivated and included because of "random" factors. That's OK, but we need to know.

Row 106. Apparently also *Larix sibirica* was present during early Holocene in central Sweden. Kullman, L., 2008 Early postglacial appearance of tree species... *QSR* 27, 2467-2472

Row 113. I interpret Table S1 as it is six lakes

Fig 2A. As no levels are analysed below 10.3 kyr in central Sweden and no younger than 10.5 in southern Sweden the study says nothing about spruce occurrence during those times. This needs to be discussed. It should also be shown in the diagram (e.g. remove the y-axis for those sections), as presented now you can get the impression spruce was absent although there is no evidence in either direction. The same is valid for the sites outside Sweden.

Row 117. Replace "east of Finland" with "Russia"

Row 118. Change to "southeast Sweden"

Paragraph starting with row 135. There are numerous mismatches between the text and figure 4, see comments below.

Row 138 and fig 4. Since the text refers to the frequencies for haplotype A, the figure should have this haplotype (blue) at the bottom. As now it is difficult to read the frequencies of the haplotype.

Row 140. Not clear which the "two other populations" are referred to here. There are five other populations in Dalarna. Or does this only refer to forest trees? Something does not make sense here.

Row 141. There are three forest populations in Jämtland, why not referring to Snasahögarna here? (by the way, the correct name is Snasahögarna, not Snåshögarna)

Row 143. Again, a mismatch? Is the "unnamed tree" Drevfjället in the figure?

Row 144 and 146. All of them except two (Sonfjället forest and Gettryggen clonal) show both haplotypes according to the figure! Hence the text is not correct.

Row 145. Is the unnamed tree Lill-skarven in the figure?

Row 151. As above, the oldest wood is around 13,000 years.

Row 154. "In these two regions, spruce trees remained present throughout the Holocene". There are no evidences spruce was present in southern Sweden during early and mid-Holocene, from this or other studies.

Row 171-173. That spruce was present in Finland until early and mid-Holocen, what is that based on? Just one finding from Finland before 6 kyr.

Row 176-178. It disappeared much earlier than late Holocene from much of northeast USA. See Lindbladh, M, Oswald, W, Foster, D, Faison, E, Hou, J & Huang, Y (2007) A late-glacial transition from *Picea glauca* to *Picea mariana* in southern New England. *Quaternary Research* 67: 502-508.

Row 181. I think understand what is meant the by birch-pine forest, ie that also other trees were present which makes it plausible that spruce could have been there, but it needs to be spelled out.

Row 185. From the north-east I assume? (Or perhaps more correct: from the east.) When it comes to migration directions, consistency and clarity is needed!

Row 200-204. Seems a little far-fetched the borderland of the north and south clusters is exactly between the areas sampled in the study. We know nothing about all the other regions in Scandinavia. And the distance is not that large between N Dalarna and S Jämtland.

Fig 5. The figure and figure text are very confusing and does not add much (other than confusion). The grey trees are difficult to detect and there are trees of many other colors that are not defined. And difficult to see a difference between the mitochondrial and nuclear data (ie top and bottom row).

Row 224-226 Not clear which "early colonizer" this regard. The northern trees migrating west through Finland, Sweden and Norway? Or the southern migrating west to Norway?

Row 229. Should be "northern and western migration route"? A route must be the direction it is moving.

Row 265-267. A circular reasoning according to me. Too many "regenerating".

Row 276. I would like the authors to discuss the risk the forest could have been planted with trees from other proveniences. A very common technique the last 50-70 years in Sweden.

Reviewer #3:

Remarks to the Author:

The authors present a good study on a topic that has been debated for a long time. The data presented here can be of help to make new conclusions and increase our understanding of the process and patterns of immigration and establishment of Norway spruce after the last glacial. The combination of studies of modern DNA and sedaDNA is important.

The paper is well written and well argued, and lets the data speak for themselves. I very little to comment to this paper.

The data selected and presented are relevant for the question asked for the paper. It will be an important contribution.

What I find mostly missing is a discussion on the climatological and ecological conditions that existed through time (should be known from previous studies) within the selected study region(s) and how that might have affected the distribution and presence of Norway spruce, and hence also the traces of sedaDNA.

One page 6, line 153 it says that spruce established in central Sweden prior to 10 cal kyr. I am not sure I agree that you can read that out of the data. There are finds of haplotype B (spruce/pine) just before 10 cal kyr, but I think more samples should have been analysed in this time to draw this conclusion. For southern Sweden that is ok.

One of the most debated questions when it comes to Norway spruce and history is the presence in the northern parts of Norway. This is not touched upon here as the sites selected for the study are mostly in central parts of Fennoscandia. A further study of this, should go west and north.

The discussion based on Figure 5 (page 7, line 200 -, and page 8, line 222 -) is a bit short. This is one of the main findings of the study and should have been elaborated more.

Some minor, concrete comments:

Page 11, line 328 – find a better word than missingness. Maybe absence.

The map in Figure 2B can maybe be difficult to read when printed.

Figure text for Figure 3A – the color reference to the sites/regions is a bit hidden.

The green arrows in Figure 5 are difficult to see – and separate from the black arrows. What is the difference between the yellow, red, and blue trees? Should there be a reference to the modern distribution of Norway spruce?

Reviewer #1 (Remarks to the Author):

Comment#1.

The authors jump to conclusions that are not based on the data they present. The discussion seems to be hiding the claimed presence of *Picea* at 15 ka in southern Sweden. This find alone needs an extensive discussion of the implications or a rejection of the find as it is difficult to reconcile with current knowledge and the other palaeobotanical data from that site as well as the second site in S-Sweden.

I cannot recommend a publication of this manuscript.

We do not intend to hide results indicating presence of *Picea* at 15kyr in southern Sweden. While we understand the reviewers concern that our findings seem contradictory to some published findings, we want to highlight that there are research findings from three independent research groups that indicate that *Picea* was present >11-17 kyr in Scandinavia (Kullman 2002 doi.org/10.1046/j.1365-2699.2002.00743.x, Paus *et al.* 2011 doi:10.1016/j.quascirev.2011.04.010; Parducci *et al.* 2012 10.1126/science.1216043). We want to highlight also that *i*) macrofossil of spruce dating to around 12 kyr have been found north of our Southern Swedish, *ii*) there is a low probability of findings the oldest fossil (i.e., *the Signor-Lipps effect*), *iii*) sedimentary DNA is recognized as the best tool to find 'ghost ranges' in ages acting beyond that of the oldest fossils (Haile *et al.* doi/10.1073/pnas.0912510106. Also notice that there are pollen-based studies that actually support our findings (see response to comment #4).

We do not agree on the reviewer's view on how recent DNA work should be interpreted in a perspective of previous pollen studies. Nevertheless, we understand the comments and criticism regarding the remaining uncertainties and accordingly in the revised version of the manuscript we have removed the word "resolve" from the title (re-phrased the title) and in the manuscript we indicate more clearly the uncertainties that remain.

Comment #2

"Personally, I would have liked to see the material divided into two more than one manuscript presented for more specialized journals. Due to the different investigations on fossil and extant material, I find the argumentation sometimes difficult to follow."

We understand if the reviewer would prefer to see the harder genetic part of our work in a separate manuscript and published in a specialized journal. Nevertheless, we think **the strength of this study is that it is integrating modern and ancient DNA data with previous paleoecological results.** We hope that our investigation can inspire further studies based on this multi-pronged approach.

Comment #3

"Given the long history of the debate and the involvement of the senior author I find it a pity that some literature is wrongly cited or omitted."

In the revised version of the manuscript, we have corrected and added new references.

Comment #4

"I have two general problems with the interpretation of the results:

1) No evidence of a survival south of the ice: When compiling the above maps, I searched for the smallest amount of pollen evidence in Lateglacial sites south of the ice and found none. The here presented interpretation that *Picea* may have survived the LGM south of the ice seems to be based solely on the results of a species distribution model based on one or two climate model

results (Svenning et al. 2008 [Journal of Ecology 96(6) doi.org/10.1111/j.1365-2745.2008.01422.x]), which are unlikely possibilities lacking any supportive evidence.”

Reviewer #1 claims that he searched for the smallest amount of pollen evidence in Late Glacial sites south of the ice and claims he found none but going through the pollen evidence in Late Glacial sites does not help understand where the spruce may have survived the LGM. Instead, one should go through LGM sites. Probably this was a typo, or a misunderstanding, however, as spruce trees near their climatic threshold does not produce pollen, i.e. reproduce by cloning (Edwards, *et al.* 2014 <https://doi.org/10.1111/geb.12213>) the point raised by the reviewer is not a strong argument. In addition, we think it not fair to criticize us for not providing supporting information besides bioclimatic models, as we are the first to report empirical evidence in support of the models (Parducci *et al.* 2012 10.1126/science.1216043).

Nevertheless, if we consider pollen evidence in Lateglacial sites, and if we consider part of the Lateglacial the period spanning between 14 and 11.7 ka BP, **results from pollen analyses at 11 sites in Skåne in southern Sweden** (Nilsson 1935; in German) **clearly show spruce pollen in this area in this period** (see table 1 below) (Lindbladh 2004 *Svensk Botanisk Tidskrift* 98:5). **Spruce pollen is found in surprisingly large quantities even in sequences from Götaland (southwest Sweden) from the late glacial period**, just in connection with the ice retreating (Lindbladh 2004). This was noticed already by Lindquist in 1948, who showed that most of these sites are located either in the southernmost part of Sweden (Scania) or close to the coast in other parts of Götaland, and Nilsson (1935) contributed the most sites (table 1 below). **Nilsson found spruce pollen in proportions of up to 11% from Older Dryas** (more than 14,000 years ago) **at all sites where this period was represented**. Based on these data, as well as the fact that spruce together with birch today form the forest border in the tundra in northern Europe, Lindquist even suggested that ‘spruce was one of the first trees to colonize the land in southern Götaland’ (Lindbladh, 2004). Our new ancient DNA data from southern Sweden strengthens this suggestion.

It is therefore possible that there were few spruce individuals sparsely distributed, likely stunted trees in favourable microhabitats that maintained themselves from the warmer periods before the LGM, like the trees today growing above the tree line in the Scandes. Our ancient DNA finding is new and interesting and suggest that woody taxa were more commonly present in areas where pollen might be absent. **We disagree with the reviewer that absence of previous paleoecological data can be used as an argument for rejecting our study as such view would make it difficult to advance the field forward**. We do acknowledge however the reviewer’s concern that this point needs to be more discussed, and we have now done so in the revised manuscript by adding a paragraph with an extensive discussion of the implications of our DNA results.

Table 1 from Lindbladh (2004) and Nilsson (1935).

Tabell 1. Antal nivåer från äldre dryas, nivåer med granpollen, maximala procentandelen granpollen i någon nivå, procent pollen från dominerande trädslag under perioden, pollen från värmeälskande trädslag. Sammanställning från Nilsson (1935).

Pollen analysis results from 11 sites in Skåne (Nilsson 1935). Columns are (from left): Number of levels from Older Dryas, number of levels with *Picea abies*, maximum percentage of *Picea abies* pollen at any level, dominating tree species, and percentage pollen of thermophilous species (tall = *Pinus*, björk = *Betula*, hassel = *Corylus*, lind = *Tilia*, al = *Alnus*).

	Nivåer Levels	Med gran With Picea	Max. andel Maximum	Dominerande trädslag Dominating trees	Värmeälskande trädslag Thermophilous trees
Allerums mosse	7	4	4 %	tall 30–60 %, björk 30–60 %	hassel 1–2 %
Vanstad mosse	26	25	11 %	tall ca 80 %, björk 10–20 %	lind 1 %, al 0–10 %
Åmossen	4	3	2 %	tall 40–75 %, björk 20–40 %	
Foteviken	41	41	10 %	tall 40–80 %, björk 10–80 %	lind 1 %
Ageröds mosse	3	2	1 %	tall 40–60 %, björk 40–60 %	
Getings mosse	4	4	7 %	tall 70–80 %, björk 10–20 %	lind 1 %, al 0–7 %
Nosaby-Hammarsjön	8	6	1 %	tall 60–85 %, björk 10–30 %	hassel 0–1 %
Kaffatorps mosse	18	17	5 %	tall 30–60 %, björk 30–55 %, al 5–15 %	lind 1 %
Baremosse	29	29	9 %	tall 30–90 %, björk 10–50 %, al 0–15 %	lind 1 %
Barsebäcks mosse	6?	6	6 %	tall 50–70 %, björk 20–40 %, al 0–5 %	
Stora Ellemossen	17	12	3 %	tall 80–90 %, björk 5–70 %	hassel 0–2 %, lind 0–1 %

Comment #5

“2) Age of oldest *Picea* detection in S-Sweden: My other main concern is the age of the *Picea* detection on the Kullen Peninsula, which at 15000 cal. BP reminds of the situation of Andøya. The first Birch macrofossils detected at that site date to about 14500 cal. BP (after the Bölling warming), which is also very early considering that the earliest finds in Denmark date to about 13700 cal. BP (Mortensen et al. 2014 [DOI 10.1007/s00334-014-0433-7]). Pushing the earliest occurrence of *Picea* in southern Sweden to before the Bölling warming and into a landscape completely bare of other trees is a significant finding that deserves a critical discussion as it has implications. The Kullen Peninsula was covered by ice during the LGM so here we are not talking about a survival of *Picea* but the spread of the tree to that place after deglaciation and before the Bölling warming.

With regards to the *Picea* detection in the core from Hässeldala, I am also surprised that an earlier publication on ancient DNA from a core of that site did not detect any *Picea* or *Pinus* DNA for the Lateglacial or Early Holocene (Parducci et al. 2019)

doi.org/10.3389/fevo.2019.00189). At this site earliest tree birch evidence seems to be dating to 13300 cal. BP, which is more in line with known regional vegetation history. Also here *Picea* would have established before *Betula* or other boreal trees, which needs a discussion.”

We would like to highlight that the claimed discrepancy between pollen and DNA is not correct (see our reply to comment #4) and that Atteköpsmosse (SWS site in our paper) is not located in the Kullen peninsula but on the coast north of this peninsula and more specifically on Hallandsåsen at 180–175 m a.s.l. (latitude 56°23'N, longitude 12°51'E). We understand however, part of the reviewer's concerns relative to this region and agree that it seems strange that spruce migrated before *Betula*, but this could again be an effect of method detection (i.e., we used spruce primers and did not search for birch) and the unlikely scenario of finding the oldest fossils as discussed under comment #1. Again, **preservation of DNA in sediments may have a greater chance of detecting spruce from a larger area (catchment size) and become preserve for long** (extracellular DNA binds efficiently, the earliest data for *Betula* macrofossils does not mean that the species was absent before.

With regards to the *Picea* detection in the core from Hässeldala, we understand if the reviewer is not familiar with the different genetic approaches used in molecular studies, but **the fact that we found no spruce in the shotgun DNA data from Hässeldala (SES site) is not surprising for us** (Parducci et al. 2019 doi.org/10.3389/fevo.2019.00189). Shotgun sequencing of environmental samples is a new technique (<10 yr old). More targeted metabarcoding methods, as we used previously (with no detection of spruce), are biased towards more abundant amplicons (common plant taxa) preventing the low abundant DNA molecules (rarer taxa) to be sequenced and detected. **For this reason, in our current study, we used a third and more efficient approach for single species detection**, which targets a multi-copy locus specifically designed for our organism of interest (Norway spruce) and this can explain why we found spruce DNA now but not before. Analysis of ancient DNA is a rapidly advancing field, and advances in methodology allow new discoveries. Both points are now addressed in the discussion of the revised manuscript.

Comment #6.

L. 44-47: The pollen diagrams reviewed in the studies 5,6 are much of the same as used in studies 7,8. There are few new (last 20 years) palaeoecological studies available from S-Sweden and no overview using the newer data. Study 8 is not showing early *Picea* pollen in S-Sweden.

We moved the reference earlier in the sentence and we talk about *Picea* in the Baltic region during the YD, and therefore, east of the ice sheet, which is mentioned in reference Giesecke, T., Brewer, S., Finsinger, W., Leydet, M., & Bradshaw, R. H. W. Patterns and dynamics of European vegetation change over the last 15,000 years. *Journal of Biogeography* 44(7), 1441–1456 (2017). <https://doi.org/10.1111/jbi.12974>.

L. 53: Why are you here not using the oldest published date of Kullman (2002), which you do cite?

The reviewer is correct, and we agree that we can be more exact in the dating and reference to Kullman.

L. 55: Based on your earlier age in this sentence the time gap would be much shorter than 10,000 years. Also this is a very interesting question (see also Giesecke 2004 <https://www.divaportal.org/smash/get/diva2:165274/FULLTEXT01.pdf>) but you are not addressing it later.

We probably had the earlier date of Kullman in mind when writing the 10,000 ky. Also, it is true that we never got back to the proposed question further in the manuscript. This is now corrected.

L. 58: Please be more specific of what you want to discuss here as an “establishment prior to deglaciation” is problematic as the tree is not growing on ice.

We agree and changed the text to “directly after the deglaciation”.

L. 99: You don't provide evidence for a survival near the southern and eastern margin of the ice. I much question the survival in the south if the ice.

We agree and we have changed the ending of the last paragraph.

L. 118: See general comment. These findings need a more extensive discussion.

We agree and we have added a new paragraph to the discussion.

L. 151: The cited publications claim older ages for the presence of Picea in the mountains.

We agree and the point has been corrected in the text.

L. 154: Which regions do you refer to here – central and southern Sweden? If so what is the evidence for continued occurrence of Picea in southern Sweden?

We refer here to central Sweden. We don't have any records for southern Sweden for most of the Holocene, so we cannot say anything about southern Sweden in the Holocene. We clarified the region we talked about in the text.

L. 156: The reference "13" is questioning the survival of trees in Norway related to the finds by Kullman(2002) in the mountains.

We changed the citation to Lemdahl and Berglund (2005).

L. 157-158: I am not disputing the presence of the tree during these times. Nevertheless, the fact that you so readily find it in most lakes is surprising as pollen evidence is very sparse in samples dating to the Early Holocene at most sites and if the tree was standing near a lake so that its DNA is preserved also the pollen should be there. I investigated 4 Sites in central Sweden scanning pollen samples for Picea pollen in Early Holocene samples and found hardly any (Giesecke 2005. doi.org/10.1016/j.quascirev.2005.03.002). So were you simply luckier than I or can you explain this mismatch?

See response to comment #4, highlighting that there is indeed other researcher that have found pollen from this time-period.

L. 169: Which "refugia east of the Weichselian ice sheet" do you refer to here?

We mean the refugia in European Russia, specifically the trees closed to the ice sheet, because trees in that area today do not have this variant anymore. So likely the trees were replaced during the Holocene. We clarified this in the text.

L. 176-178: The geographical setting in North America is completely different so you should elaborate which parallels you see exactly.

We agree with this comment and removed this sentence from the text.

L. 181: Yes, but trees were subsequently expelled from these locations so what is the relevance here.

We think the relevance is that the short haplotype (A) was present at that period, prior to the disappearance of the trees which indicates that this variant was present much further east than it was believed in Parducci *et al.* 2012 10.1126/science.1216043.

L. 183: The study did not focus on Picea and the results regarding Picea are very questionably and not at all supported by any palaeoecological evidence.

We agree that ecological niche modelling studies without palaeoecological evidence are not a prove of presence of trees. However, the study implies that local microclimates may have sustained spruce

populations in areas for which there is no palaeoecological evidence today. The study Svenning *et al.* 2008, does not claim trees were there, it is merely suggesting they potentially could be there.

L. 222-223: I don't see the logical connection of the argument here and I much dispute the survival from the south for which there is no evidence.

We agree and have now reformulated some key sentences in the paragraph to convey that a southern refugia is suggested but we don't have direct evidence.

L. 227: I don't see what you resolve. The existence of the tree in Scandinavia during the early Holocene was already considered by Giesecke and Bennett (2004) and you don't present evidence for the survival near the margin of the LGM ice.

We agree and we removed therefore the word 'resolve' from the title and in main text. We hypothesized, but not independently confirmed, that trees may have survived south. As we found no unique genetic material in the clonal trees of the central Scandes we exclude the importance of a western refugia. We now improved the formulation of the paragraph.

Reviewer #2 (Remarks to the Author):

Comment #1

Spruce is the most important tree species in Fennoscandia no matter what aspect: Economy, recreation, biodiversity, etc. Yet, its history is still shrouded in obscurity after more than 100 years of investigations and debate. The study is novel and confirms a very early presence of spruce in Scandinavia, which certainly is good enough for justifying publication in Nature Communication, and I recommend it to be accepted. However, it does not, as claimed, solve the question about spruce origin and migration routes and it needs a major revision before being accepted.

As I am not a geneticist, I cannot assess the quality of the genetic part in the study. I assume another reviewer with that expertise has been assigned. However, I know from discussions in the scientific community that there could be a large risk for contamination. The DNA is small, to say the least. I have no opinion in the matter, although with my novice perspective their arguments in the supplementary information seem convincing. Anyway, the text regarding this subject should not to be hidden there but to be discussed in detail in the main manuscript – and scrutinized by expertise in the field.

We thank the reviewer for acknowledging the importance of our study. We also think that it is very important that **our DNA results are scrutinized by expertise in the ancient DNA field**. However, due to its lengths and technicality, we think that the contamination discussion should remain in the supplementary information, but we agree with the reviewer that it should be better emphasized in the manuscript and now added a short paragraph on this in the main text.

Comment #2

The very same wetland analysed here for ancient DNA (SWS) was also analysed by Wohlfarth *et al.* (2018) for among other things pollen, macrofossils, hydrogen isotopes and chironomids. In contrary to Nota *et al.* their study does not at all support the presence of spruce at the time. Apart from finding no physical evidence of the tree, their reconstruction of vegetation and climate showed anything but a spruce-friendly environment. According to them, it was relatively warm 15,000 years ago, but there were no trees. Instead, it was an open arctic landscape dominated by herbs, grasses and dwarf shrubs. I would like the authors to comment on that and discuss possible reasons behind the (large) discrepancy between the studies.

We understand well this concern as explained above in our reply to a similar comment made by reviewer #1. In our opinion DNA results should not be dismissed because of lack of previous palynological evidence, because this is the reason why we are using new modern DNA-based methods instead of relying only on traditional pollen data which is known to have some limitation. **Our approach is to try to integrate modern and ancient DNA data with previous paleoecological results**, as the different approaches are often complementary and help each other; but of course, it happens also that the two are contradictory and results controversial. It is important to recognize that neither approach is definitive. We now address this point better in our text and discussed it thoroughly.

Comment #3

Row 74-76 and 130-132. “Each if these clones...likely originating from a single ancient individual”. Connected or not is crucial for this study. It has been questioned if the wood in the soil actually was connected to the living “Old Tjikko” tree. See <https://kimmerer.com/how-old-is-old-tjikko/> and references therein. If not, the same could be the case for all clonal trees in the study. I would like the authors to comment on the possible consequences for the study if they are not connected. If the living tree is of much younger origin than the wood in the soil, and sexually generated, it is not surprising the living trees share DNA with trees in the surrounding forest, and many of the conclusions in the paper must be revised.

We acknowledge that there is uncertainty and debate about the direct connectivity between the clonal trees still living today and the fossils found directly under the tree. There is strong evidence that trees have continuously occupied the area during the Holocene, therefore, from a genetic point of view, genotypes (SNPs) will be retained in the population during sexual generation, because trees with similar genetic lineages are mixing together in a micro habitat. At most, recent admixture would be observed between clonal trees and forest trees arriving later if they mixed recently, and contained different distinct genetic lineages. Whether the clonal trees are relatively young, and a result of sexual reproduction will not change the conclusions of the manuscript. For a more in-depth answer to this question see below*.

*Our ancient DNA results show that spruce was present early and throughout the Holocene in Jämtland. The question whether the old clonal trees could be survivors and contributed to the next generation is only relevant when we accept that they are indeed ancient and originate before the massive spruce expansion in the late Holocene. If they are ancient, there is no evidence that they are different from the spruce forest. If they are more recently established (late Holocene) they must have replaced the population established in the early Holocene, in this scenario too, the potential survivors cannot be excluded, but they did not contribute to the recolonization of Fennoscandia.

The genetic link between the dated macrofossils and clonal spruce trees present today such as Old Tjikko is an ongoing debate. Our study, too, cannot show direct evidence of genetic connectivity, and there is no way to do this with convincing evidence (DNA in macrofossils is too little, contaminated and damaged for being securely analysed), however indirectly we strongly suggest so, as haplotype A is present throughout the Holocene based on our sedaDNA results. This result rejects a scenario of complete replacement of the populations during the Holocene because this haplotype is not present outside Sweden and Norway, while in the Scandes is nearly fixed in some populations.

The question raised by the reviewer is about the impact of the conclusions drawn in our study if the sampled clonal trees above the treeline are young and not connected to the early Holocene colonisers. We argue that our conclusions would not change and do not need to be revised based on the following lines of reasoning in six potential scenarios:

- A) *Direct genetic link between dated macrofossils and trees – distinct western glacial refugia.* In this case, the genetic variation present in the earliest established trees would not have changed and not be admixed with the variation of trees arriving later in the Holocene. Clonal trees would be grouping away from the surrounding forest. A signal would have been present, and we would have suggested that they originated from a separated glacial refugia but importantly did not contribute to the recolonization of Scandinavia.
- B) *Direct genetic link between dated macrofossils and trees – eastern glacial refugia.* In this case, the genetic variation present in the earliest established trees would not have changed and not be admixed with genetic variation of trees arriving later in the Holocene. Clonal trees and forest trees would share a similar genetic history and would likely still be admixed between the northern and southern clusters, because the area is a contact zone, and early colonisers could have arisen from admixed populations. In this case there is no signal between clonal and forest trees, as detected in our study, and we therefore conclude a likely eastern origin and that there is no separated glacial refugia, which importantly did not contribute to the recolonization of Scandinavia. However, it is not possible to exclude western refugia because genotypes in the refugia are the same.
- C) *No direct genetic link between dated macrofossils and trees but continuous presence – distinct western glacial refugia.* In this case, the genetic variation present in the earliest established trees would be maintained and be at most admixed with genetic variation of trees arriving later in the Holocene. A signal would have been present and we would suggest that they originated from a separated glacial refugia, but importantly, we would still conclude they did not contribute to the recolonization of Scandinavia.
- D) *No direct genetic link between dated macrofossils and trees but continuous presence – eastern glacial refugia.* In this case, the genetic variation present in the earliest established trees would be maintained and be at most admixed with genetic variation of trees arriving later in the Holocene. Clonal trees and forest trees would share a similar genetic history and would likely be admixed because the area is in a contact zone. In this case there would be no signal between clonal and forest trees as detected in our study, and we would conclude eastern origin and that there is no separated glacial refugia which importantly did not contribute to the recolonization of Scandinavia. But we could not exclude the possibility that few trees survived in the west because genotypes in the refugia are the same.
- E) *No direct genetic link between dated macrofossils and trees without continuous presence – distinct western glacial refugia.* In this scenario there were time periods where the clonal populations were not present and were replaced one or multiple times during the Holocene. In this case, the spruce populations would be young, and resemble the observed data, with no separation of the clonal and surrounding forest. It would not be possible to detect the distinct western refugia because the connectivity is gone.

- F) *No direct genetic link between dated macrofossils and trees without continuous presence – eastern glacial refugia.* In this scenario there were time periods where the clonal populations were not present and were replaced one or multiple times during the Holocene. In this case, the spruce populations would be young, and resemble the observed data, with no separation of the clonal and surrounding forest. Not possible to exclude western refugia because genotypes in the refugia are the same.

Scenarios A and C can be rejected because there is no signal of a distinct western genotype. Scenarios B and D would both produce the results observed in our study, and both would lead to the same conclusions: early colonisation from east or east and south with no rejection of a western refugia because trees might have belonged to the same two widespread genetic clusters. Scenarios E and F would produce the genetic structure observed in our study with nuclear markers, but it would not agree with the distribution of the mitochondrial haplotype A, because there would be no source area for this haplotype.

Therefore, according to our data B and D are the most likely scenarios and since we believe there is genetic continuities between dated macrofossils and trees, scenario B is in our view more likely than D. However, since both are possible, it is even possible a combination of the two. Nevertheless, the most important thing is that **our conclusions do not differ depending on true genetic connectivity between fossils and trees, or genetic continuity through sexual reproduction during the Holocene.**

Comment #4

It must be shown in diagram 2 which time periods that were sampled (e.g. remove the y-axis for the sections not sampled), as presented now you can get the impression spruce was absent although there is no evidence in either direction. As now, it is very difficult to interpret the evidences for spruce (haplotype A) presence or non-presence, and it is difficult to follow the authors discussion.

Thank you for pointing us this issue. We did placed dots on the y-axes which indicate the places samples were taken but to make this clearer we used now the grey colour to shadow the part of the core which have not been sampled and analysed.

Comment #5

Row 192-an onwards. What's the evidence that today's local forest originates from the east? I don't follow the logic here. As the clonal trees are more similar to the surrounding forests that came later than to other clonal trees in the mountains, it seems hard to believe there have been no contact between the new forest and the living (clonal) trees. How else can they be so similar, when they have been separated for thousands of years and thousands of kilometres? This raises the question whether the living trees (Old Tjikko for instance) actually are clones originating from the old wood in the soil. The study does not present convincing evidences or even indications, as claimed in rows 226-230, that spruce arrived to central Sweden early from the south and north. Neither does it explain the origin of the very early (up to 13,000 year old) spruce wood in the mountains.

Extensive previous molecular studies of contemporary spruce in Fennoscandia (all cited in the manuscript) show that there are two distinct genetic clusters in Fennoscandia, one in the south, and one in the north, with a contact zone in central Sweden. The trees in southern Sweden have the same genetic makeup as the ones in southern Finland and European Russia, and the trees in northern Scandinavia are similar to the ones in northern Finland, and the northern areas of European Russia. The trees in the Baltic region group close to the Fennoscandian ones but are more distinct. There is not enough evidence to suggest that spruce migrated from west to east, therefore, we conclude that the modern forest that exists in the area today arrived from the east, either in the early or in the late Holocene. With detecting spruce so early in southern Sweden, there might have been a source of trees migrating north to the Scandinavian mountains. That the trees in the valleys are so similar to the nearby located clonal trees shows indeed that there has been a connection in the past (originating from the similar source population except at different times). The important question is whether this happened thousands of years ago (they have the same origin) or are they are similar because of recent contact.

We know that spruce is very slowly evolving, as all conifers in general, and with high and long-distance geneflow the populations are maintaining low genetic differentiation. For example, the trees in the Alps have been separated for millions of years, yet the amount of genetic differentiation with the Fennoscandian trees remains low. There is no evidence that in long-lived, slowly evolving conifer species, genetic variation is modified by glacial cycles and therefore trees in southern Sweden are genetically similar to those in Russia because they have not been separated long enough. This also applies to clonal trees that originated from populations in the east and are still the same, even if there was no geneflow in the last 8000 years. It is true that the levels of admixture are the same in the valleys and the clonal populations, however, this admixture could have happened in the early Holocene when the trees established. Our conclusion is that it is true that a single modern colonisation would result in the same observation, but it would imply that the early Holocene spruce trees disappeared before the new trees arrived and in addition that they established above the treeline.

Minor comments

Row 31. It is not north-east, but rather east.

We corrected this point.

Row 44-47. But there is one study with pollen that was considered to come from local origins: Segerström, U. and H. von Stedingk, Early-Holocene spruce, *Picea abies* (L.) Karst., in west central Sweden as revealed by pollen analysis. *Holocene*, 2003. 13(6): p. 897-906

Thank you for pointing this and accordingly we changed the sentence and add this information.

Row 49. The oldest spruce wood in the Kullman publication is older than that: around 13,000 cal BP (Beta-121826 11,020 ±90 *Picea abies* Stem)

We forgot to convert the radiocarbon data to the yr BP. We apologise for this mistake and corrected the date.

Row 54. “until” better than “only” according to me

We corrected this point.

Row 59. The directions here are problematic, does an eastern route mean it came east or migrated to the east? I assume it here means: “migrated to the west”, or “a western migration route”. In several cases in the text it is unclear whether the direction regards from where the spruce migrated or if it is in which direction it migrated to.

We agree that there was some confusion, and we clarified all the places where it was confusing

Row 61-62. Well, if it did come from the south, it left quite significant traces in the pollen record: Check table 1 in:

Lindbladh, M., När granen kom till byn – några tankar kring granens invandring till södra Sverige (The immigration of spruce to southern Sweden). *Svensk Botanisk Tidskrift*, 2004. 98: p. 249-262.

We thank the reviewer for this information, and we now better acknowledge this point in our discussion.

Row 63-64. He first brought up the idea in 2002, in publication no 10 here.

We corrected this point.

Row 67. kyr, not yr.

We corrected this point.

Row 71. Citation mark missing.

The missing citation mark is now added.

Row 88-89. What if the DNA come from pollen? It is unlikely large pieces of wood end up on the lake bottom or on a mire, and pollen can make up pretty large quantities.

Large pieces of wood are not the only source of DNA in sediments. There are many studies suggesting that pollen contribute little or not at all to the DNA signal in lake sediments (Parducci *et al.* 2017 doi: 10.1111/nph.14470). It is not yet known where the DNA is coming from exactly, and it depends also on the plant species in question, but most likely what contributes most to sedDNA are small plant parts like needles, seeds or fruits that end up in the lake and degrade, or even from water runoff in the catchment that transports DNA from roots in the catchment. DNA is then released in the sediments (as the macrofossils dissolve) but protected as extracellular DNA when it binds to the mineral particles (Parducci *et al.* 2017 doi: 10.1111/nph.14470). In bogs and mires, roots might be also a source of DNA or fallen needles.

Row 92 and onwards. Somewhere here the choice of sites needs to be discussed. The mountain sites in Sweden are easy to motivate due to the findings of ancient wood. The other sites are perhaps less easily motivated and included because of “random” factors. That’s OK, but we need to know.

Thank you for noticing this. We have now better motivated the site choice in the materials and methods section.

Row 106. Apparently also *Larix sibirica* was present during early Holocene in central Sweden. Kullman, L., 2008 Early postglacial appearance of tree species... QSR 27, 2467-2472

Thank you for noticing this, which is interesting, and we missed it. It is difficult to check if *Larix* could have the short haplotype because the local tree populations in Fennoscandia are no longer there. That *Larix* contained the short variant seems highly unlikely however, since haplotype A is due to a deletion mutation occurred in the most common variant haplotype B present in most of the other sister species to spruce (Parducci *et al.* 2012). We have added this to the main text.

Row 113. I interpret Table S1 as it is six lakes

The reviewer is correct, and we acknowledge that there was some confusion, because one of the sites is actually in Norway. We changed it to six lakes.

Fig 2A. As no levels are analysed below 10.3 kyr in central Sweden and no younger than 10.5 in southern Sweden the study says nothing about spruce occurrence during those times. This needs to be discussed. It should also be shown in the diagram (e.g. remove the y-axis for those sections), as presented now you can get the impression spruce was absent although there is no evidence in either direction. The same is valid for the sites outside Sweden.

Thank you for pointing us this issue. We had originally placed dots on the y-axes to indicate the places the samples were taken but to make this clearer now we used the grey colour to shadow the part of the core which have not been sampled and analysed.

Row 117. Replace “east of Finland” with “Russia”

Replaced.

Row 118. Change to “southeast Sweden”

Changed

Paragraph starting with row 135. There are numerous mismatches between the text and figure 4, see comments below.

We think there are no real mismatches but probably a misunderstanding or confusion between the concepts of 1) haplotype of an individual tree and 2) haplotype frequency within a population (displayed in figure 4). We made now the distinction clearer in the figure legend and by adding number of individuals above the bars.

Row 138 and fig 4. Since the text refers to the frequencies for haplotype A, the figure should have this haplotype (blue) at the bottom. As now it is difficult to read the frequencies of the haplotype.

Thank you for pointing this out. We have changed the colour of the bars.

Row 140. Not clear which the “two other populations” are referred to here. There are five other populations in Dalarna. Or does this only refer to forest trees? Something does not make sense here.

This is now corrected, indeed there are more than two other populations.

Row 141. There are three forest populations in Jämtland, why not referring to Snasahögarna here? (by the way, the correct name is Snasahögarna, not Snåshögarna)

The spelling mistake in Snasahögarna is now corrected. We made it clearer which populations we meant.

Row 143. Again, a mismatch? Is the “unnamed tree” Drevfjället in the figure?

Not exactly, since the bars in figure 4 are not individual trees, but they represent about 20 individuals collected in the same geographical area. There are two radiocarbon dated trees in Härjehågnå which are unnamed. We will make it clearer that the figure shows the frequency at which the haplotype occurs in a population and not a single individual.

Row 144 and 146. All of them except two (Sonfjället forest and Gettryggen clonal) show both haplotypes according to the figure! Hence the text is not correct.

We apologise for not being sufficiently clear enough in our formulation and we now correct the text accordingly. The trees with a date and a name are only few among those that make up a population. The text in 144-146 describes the haplotype (genetic variant) of these particular trees. The results presented above show instead the proportion of trees in the population that have a certain haplotype (single trees have either A or B and cannot have not both).

Row 145. Is the unnamed tree Lill-skarven in the figure?

No, the Lill-skarven population is composed entirely of clonal trees and includes some individuals for which macro fossils have been dated.

Row 151. As above, the oldest wood is around 13,000 years.

Now corrected in the text.

Row 154. “In these two regions, spruce trees remained present throughout the Holocene”. There are no evidences spruce was present in southern Sweden during early and mid-Holocene, from this or other studies.

The reviewer is correct, and we actually meant within the two regions Jämtland and Dalarna. We corrected the text to make it clearer we do not mean southern Sweden.

Row 171-173. That spruce was present in Finland until early and mid-Holocene, what is that based on? Just one finding from Finland before 6 kyr.

We know from pollen already that spruce was present in Finland before 6 kry BP but in our study, we found that some of these trees had the short haplotype A. We mean that the haplotype A disappeared but we cannot make a statement that spruce disappeared completely. We clarify this in the text.

Row 176-178. It disappeared much earlier than late Holocene from much of northeast USA. See Lindbladh, M, Oswald, W, Foster, D, Faison, E, Hou, J & Huang, Y (2007) A late-glacial transition from Picea glauca to Picea mariana in southern New England. Quaternary Research 67: 502-508.

We corrected this in the text.

Row 181. I think understand what is meant the by birch-pine forest, ie that also other trees were present which makes it plausible that spruce could have been there, but it needs to be spelled out.

We added this to the text.

Row 185. From the north-east I assume? (Or perhaps more correct: from the east.). When it comes to migration directions, consistency and clarity is needed!

We thank the reviewer for pointing out this issue and we have clarified all the directions to make them more consistent and clearer in the text.

Row 200-204. Seems a little far-fetched the borderland of the north and south clusters is exactly between the areas sampled in the study. We know nothing about all the other regions in Scandinavia. And the distance is not that large between N Dalarna and S Jämtland.

Thank you for pointing this out. We do not claim that we have sampled the complete contact zone. Li et al 2020 show that the zone is small and there is no variation further north and south.

Fig 5. The figure and figure text are very confusing and does not add much (other than confusion). The grey trees are difficult to detect and there are trees of many other colors that are not defined. And difficult to see a difference between the mitochondrial and nuclear data (ie top and bottom row).

Now we improved the legends and the colours hoping the reviewer find the figure better readable.

Row 224-226 Not clear which “early colonizer” this regard. The northern trees migrating west through Finland, Sweden and Norway? Or the southern migrating west to Norway?

Either of these routes is possible, and we are not able to make a distinction based on our results.

Row 229. Should be “northern and western migration route”? A route must be the direction it is moving.

We are not sure we understand this comment. The directions are from the perspective of the Scandes, so they are coming from the south and east. If we are talking from the perspective of the refugia, they instead moved west and north. Nevertheless, the sentence is now removed from the paragraph because of comments received from reviewer #1.

Row 265-267. A circular reasoning according to me. Too many “regenerating”.

Thank you for pointing this out, however we are not sure we understand where the circular reasoning is. On the other hand, we see there are too many repetitions of the word “regenerating” in the sentence, and we therefore reformulated It.

Row 276. I would like the authors to discuss the risk the forest could have been planted with trees from other proveniences. A very common technique the last 50-70 years in Sweden.

This is always a risk, but we took precautions while sampling and we never found indications that trees from other provenances were planted in the study areas. Moreover, we find a clear sign of genetic isolation by distance and congruence DNA results which did not indicate the trees were planted. In addition, we think that the signal would not be so disturbed if trees were moved around in the last 50-70 years and imported from different regions. We added a sentence about this in the materials and methods section.

Reviewer #3 (Remarks to the Author):

Comment #1

The authors present a good study on a topic that has been debated for a long time. The data presented here can be of help to make new conclusions and increase our understanding of the process and patterns of immigration and establishment of Norway spruce after the last glacial. The combination of studies of modern DNA and sedaDNA is important.

The paper is well written and well argued, and lets the data speak for themselves. I very little to comment to this paper.

The data selected and presented are relevant for the question asked for the paper. It will be an important contribution.

We thank the reviewer for acknowledging the importance of our study.

What I find mostly missing is a discussion on the climatological and ecological conditions that existed through time (should be known from previous studies) within the selected study region(s) and how that might have affected the distribution and presence of Norway spruce, and hence also the traces of sedaDNA.

We agree with this important comment, and we now added a discussion about this point in the manuscript.

One page 6, line 153 it says that spruce established in central Sweden prior to 10 cal kyr. I am not sure I agree that you can read that out of the data. There are finds of haplotype B (spruce/pine) just before 10 cal kyr, but I think more samples should have been analysed in this time to draw this conclusion. For southern Sweden that is ok.

We agree, as based on our data alone we cannot make this conclusion, though we have positive presence from at least 9800 cal. yr BP. We have now corrected this point in the text.

One of the most debated questions when it comes to Norway spruce and history is the presence in the northern parts of Norway. This is not touched upon here as the sites selected for the study are mostly in central parts of Fennoscandia. A further study of this, should go west and north.

Thank you for pointing this for us and we agree that the question is much debated too, however in our study we were mainly interested about the origin of the clonal population in the Scandes.

The discussion based on Figure 5 (page 7, line 200 -, and page 8, line 222 -) is a bit short. This is one of the main findings of the study and should have been elaborated more.

We agree with the reviewer comment and we discussed this point more extensively now and discussed more its implications.

Some minor, concrete comments:

Page 11, line 328 – find a better word than missingness. Maybe absence.

In the context of nuclear SNP data, the term ‘missingness’ is normally used, but we agree that it can be unclear for a broader audience, and we now changed it to ‘absence’.

The map in Figure 2B can maybe be difficult to read when printed.

We agree and we will of course change the size of the map if the manuscript is accepted and when it will be edited for printing.

Figure text for Figure 3A – the color reference to the sites/regions is a bit hidden.

Thank you for pointing this for us. We corrected and we hope that now the colour legend is clearer.

The green arrows in Figure 5 are difficult to see – and separate from the black arrows. What is the difference between the yellow, red, and blue trees? Should there be a reference to the modern distribution of Norway spruce?

Thank you for pointing this issue and we made now clearer with extra labels that the tree colouring is identical to the colouring of the data presented in figure 3 and 4 and we added a reference the modern distribution of Norway spruce.

Reviewers' Comments:

Reviewer #1:

Remarks to the Author:

I apologize for the strong tone in the initial comments on this manuscript. That text was meant for the editor and not for the authors. The authors have indeed toned-down the manuscript, which now reads more balanced. While the manuscript as it stands makes sense, even if I disagree with some detail, the Picea amplifications dating to 14.7 ka need a critical discussion, for which there is no room in the current manuscript. Earlier Lateglacial claims of Picea finds have always resulted in controversial discussion and not all the presented evidence is undisputed. All other claims place the early evidence into a time where a warmer Lateglacial climate promoted the spread of trees into Scandinavia, giving a chance for long distance dispersal events. This new evidence suggests that we had Picea growing at a site that was previously glaciated within less than 100 years (dating uncertainty) of the onset of the Bølling warming. Let's assume this was not a single tree but a population, then we need time for the population to build up. Picea must have gotten there via seed dispersal. As I stated earlier, I am not aware of any evidence suggesting the survival of Picea south of the LGM ice. Assuming a source population in the east, we need dispersal events of more than 1000 km and a population build up within decades, which is unrealistic. The alternative would be suggesting that trees were on the move already before the Bølling warming. The authors often take parallels to the clonal growth of Picea in the mountains surviving without the production of pollen. Well this would not be a possible parallel here as the establishment of a population needs seeds which means there was pollen involved. Evidence for the growth of Picea in an otherwise woodless southern Scandinavia has thus many consequences and implications and it is therefore important to solidify such a find with a critical discussion. Without this it will be difficult to use this information. Therefore, I would still suggest removing the two Lateglacial studies from the current manuscript and publishing them separately, allowing for a critical discussion of that evidence and its implications, not just for Picea, but also our understanding of the vegetation response to Lateglacial climate change.

Detailed comments:

Line 49-52: Please don't use "early pollen records" and delete "event". As stated earlier the pollen diagrams used for 8 and 10 are the same. Moreover, several co-authors of this study know about the problem of redeposited pollen in the basal sediments of lakes depicted in 8.

Line 56: Yes, of course if you are in a vegetation producing lots of pollen. See my simulation in Gaillard et al. (2008). *Vegetation History and Archaeobotany*, 17(5), 419-443.

Line 64 ff: It would be good to sharpen the conflicting hypothesis. As you address two different questions in this manuscript (the origin of the old trees above the treeline and the more general recolonization of Scandinavia) arguments seem to get mixed here.

Line 192: As stated above, Lateglacial Picea pollen is often present in the basal sediments of lakes together with pollen types from thermophilous trees and therefore interpreted as coming from older interstadial or interglacial material and redeposited. In this connection it would be interesting to know if this redeposited material may also still contain aDNA that could be amplified. The amplification of aDNA markers dating to 42 ka at Finnish sites would suggest that this possibility exists.

Line 219 ff: In the discussion of the work of Nilsson it would be important to state the pollen sum that the Picea percentage is based on. I assume that he used a tree pollen sum at that time. With hardly a tree being around most tree pollen would have been long distance transport and thus a value of 11% for Picea is not very informative. Also some assentation on the dating of the sites to around 14 ka would be useful as he was not able to independently date his findings. Assuming these finds are indicating the presence of Picea in southern Sweden, the here presented finds are still predating that by 1 ka. Also the assentation of "high July summer temperatures and continental climate throughout the Lateglacial (15-11 cal. kyr BP) in southern Sweden" is not solving my above problem that the finds presented here date to right the beginning of that warm phase. You state: "Stable soils and warm summer temperatures could have been favourable for spruce to colonize the area." Yes, but from all we know so far this is a process needs more than 100 years as implied by the dating of this find.

Line 236: The successful amplification of a DNA dating to around 43 ka suggests the possibility that

such old sediments containing the here investigated markers were redeposited in the lake sediments resulting in the detection of Picea dating to 14.7 ka. Could you comment on this?

Line 253: Nobody disagrees that Picea was present in the East - where in the East or how far north is a more interesting Question. Please be more specific.

Line 292 – 295: I know what you want to say here but readers less familiar with the debate may get lost here.

Line 304: "via water transport": It is not clear what is imagined here. - Seeds drifting on logs across the Baltic ice lake? But how to they get on top of the mountain via water transport?

Fig. 5: While you discarded the possibility of a survival of Picea south of the ice during the LGM in the text, the maps are still entertaining this possibility. So either you remove it from the maps or you discuss evidence for it in the text. I am not aware of any evidence, but may of cause be wrong. Still these maps will be used in the same way you used the maps by Svenning et al..

Reviewer #2:

Remarks to the Author:

I am quite happy with the revision and recommend the paper for publication. As stated in my first review the new findings in the study is of large importance for the scientific field. A few minor things:

Row 134. Something is wrong with the sentence starting with "As the mh05..."

Row 220. Note that Nillson's spruce pollen are from OLDEST Dryas (about 15,000-17,000 cal BP). And not from Older dryas. See reference 21.

Row 255. Connection or not between the stem and the wood is important. I have a hard time (and lack of time) to scrutinize "The more in-depth answer" in the rebuttal, but from my perspective the reasoning seems convincing. Perhaps should it be included in the appendix?

Reviewer #3:

Remarks to the Author:

Dear Authors,

Thank you for doing a very thorough revision of your manuscript. I think your replies are well argued.

Lines numbers refer to the pdf version with track changes highlighted.

Reviewer #1 (Remarks to the Author):

Comment #1 I apologize for the strong tone in the initial comments on this manuscript. That text was meant for the editor and not for the authors. The authors have indeed toned-down the manuscript, which now reads more balanced. While the manuscript as it stands makes sense, even if I disagree with some detail, the *Picea* amplifications dating to 14.7 ka need a critical discussion, for which there is no room in the current manuscript. Earlier Lateglacial claims of *Picea* finds have always resulted in controversial discussion and not all the presented evidence is undisputed. All other claims place the early evidence into a time where a warmer Lateglacial climate promoted the spread of trees into Scandinavia, giving a chance for long distance dispersal events. This new evidence suggests that we had *Picea* growing at a site that was previously glaciated within less than 100 years (dating uncertainty) of the onset of the Bølling warming. Let's assume this was not a single tree but a population, then we need time for the population to build up. *Picea* must have gotten there via seed dispersal. As I stated earlier, I am not aware of any evidence suggesting the survival of *Picea* south of the LGM ice. Assuming a source population in the east, we need dispersal events of more than 1000 km and a population build up within decades, which is unrealistic. The alternative would be suggesting that trees were on the move already before the Bølling warming. The authors often take parallels to the clonal growth of *Picea* in the mountains surviving without the production of pollen. Well this would not be a possible parallel here as the establishment of a population needs seeds which means there was pollen involved. Evidence for the growth of *Picea* in an otherwise woodless southern Scandinavia has thus many consequences and implications and it is therefore important to solidify such a find with a critical discussion. Without this it will be difficult to use this information. Therefore, I would still suggest removing the two Lateglacial studies from the current manuscript and publishing them separately, allowing for a critical discussion of that evidence and its implications, not just for *Picea*, but also our understanding of the vegetation response to Lateglacial climate change.

We thank Reviewer #1 for the comments, and we are happy that reviewer found our revised manuscript more balanced. We are also convinced that we have been able to handle all comments and hope that this version will meet all concerns.

We respect the reviewer's opinion that the finding of *Picea* DNA in Lateglacial sediments shows a poor match with the established knowledge about spruce migration and thus, should be treated with caution. Established and conventional knowledge is however predominantly based on pollen as a paleoecological proxy for the past occurrence of plants, which differs from the proxy presented in our manuscript. The use of pollen as a sole proxy for the occurrence/presence of plants can for example not explain findings of spruce macro- and megafossils dating back to the Late glacial (Kullman, 2006) or other trees like Scots pine (*Pinus sylvestris*) growing in Northern Sweden (Klaminder, unpublished data at the end of this document*) more than a millennia before pollen studies detected their arrival (Cheddadi *et al.*, 2006). We would also like to highlight that the basal age of our studied lake in southern

Sweden dates to 15.5 cal. kyr BP and provides a better estimate of the deglaciation of the area than the regional deglaciation maps that the reviewer seems to refer to. This means that spruce had ~800 years to colonise the lake's surroundings after the disappearance of the ice, and not 100 years as reviewer #1 suggests. Therefore, we see no reason to add a comment about the dating uncertainty to the manuscript.

On the other hand, removing entirely our findings from southern Sweden for the readers of our paper, as suggested by the reviewer, seems unethical to us. Nevertheless, to meet this important criticism raised by the reviewer we have now in the revised version of the manuscript as follows:

- i) We adopted the most conservative approach and avoided interpretation of any samples located at the transition between lake sediment and the underlying till, i.e., to minimize the risk of interpreting DNA from a potentially relict landscape (see response Comment #5. In the text changes were made on lines 212-216).
- ii) We added a sentence in the discussions highlighting the current need of searching for macrofossil findings in southern Sweden to support our findings (Lines 317-321).
- iii) We clearly stated that we are not presenting our result as a final proof for spruce survival south of the Scandinavian Ice Sheet and propose alternative hypotheses to glacial survival (Lines 307-311, 339-344).
- iv) We modified the legend of Figure 5 where we explain that, because we do not present solid palaeoecological evidence, we placed question marks on the trees (Lines 514-515).

Detailed comments:

Comment #2: Line 49-52: Please don't use "early pollen records" and delete "event". As stated earlier the pollen diagrams used for 8 and 10 are the same. Moreover, several co-authors of this study know about the problem of redeposited pollen in the basal sediments of lakes depicted in 8.

We agree with the reviewer, and we removed "early" from the "pollen records" and deleted "event" from the sentence (From line 77).

Comment #3. Line 56: Yes, of course if you are in a vegetation producing lots of pollen. See my simulation in Gaillard et al. (2008). Vegetation History and Archaeobotany, 17(5), 419-443.

Thank you for this comment. Segerström & Stedingk (2003), however, suggested that in forested environments even low percentages of spruce can be considered of local origin.

Comment #4: Line 64 ff: It would be good to sharpen the conflicting hypothesis. As you address two different questions in this manuscript (the origin of the old trees above the treeline and the more general recolonization of Scandinavia) arguments seem to get mixed here.

We thank the reviewer for this comment; however, we are uncertain how to follow the suggestion. We think that the two issues of origin and direction of migration are connected and cannot be studied or discussed separately.

Comment #5: Line 192: As stated above, Lateglacial Picea pollen is often present in the basal sediments of lakes together with pollen types from thermophilous trees and therefore interpreted as coming from older interstadial or interglacial material and redeposited. In this connection it would be interesting to know if this redeposited material may also still contain aDNA that could be amplified. The amplification of aDNA markers dating to 42 ka at Finish sites would suggest that this possibility exists.

We agree, and to meet this important comment about redeposition, we explain now in the paper that we avoided using the most basal sediment samples located near the transition between till and lake sediments (Lines 212-216). Nevertheless, we would like to highlight that pollen, microfossils and ancient DNA from the 42-ka interstadial is mainly found in the reworked till deposited beneath Holocene sediments and not in the Holocene sediment *per se* (Zale *et al.* 2018). This implies that interglacial deposits containing pollen and DNA fragments from the MIS-3 interstadial (ca 45 ka) were overridden during the last advance of the Weichselian ice sheet and buried at depth in glacial debris. We believe therefore, that strong inputs of DNA from MIS-3 (or older ecosystems) is highly unlikely in our case.

Comment #6: Line 219 ff: In the discussion of the work of Nilsson it would be important to state the pollen sum that the Picea percentage is based on. I assume that he used a tree pollen sum at that time. With hardly a tree being around most tree pollen would have been long distance transport and thus a value of 11% for Picea is not very informative. Also some assentation on the dating of the sites to around 14 ka would be useful as he was not able to independently date his findings. Assuming these finds are indicating the presence of Picea in southern Sweden, the here presented finds are still predating that by 1 ka. Also the assentation of “high July summer temperatures and continental climate throughout the Lateglacial (15-11 cal. kyr BP) in southern Sweden” is not solving my above problem that the finds presented here date to right the beginning of that warm phase. You state: “Stable soils and warm summer temperatures could have been favourable for spruce to colonize the area.” Yes, but from all we know so far this is a process needs more than 100 years as implied by the dating of this find.

It is true that in the past it was common to calculate the tree percentage values from the total sum of trees. Thus, if it is 11% of tree pollen and if the environment was mostly treeless, the total proportion of tree pollen may have been, around 20-40% and therefore a spruce pollen value of 11 % would not be as high as it may sound. We did acknowledge this in the paper, and we now added more words to clarify the issue relative to the pollen sum (Line 241). To meet even better the criticism raised by the reviewer, now we cited' also a paper which was criticising Nilsson's conclusions (Lines 236-239).

Regarding the dating issue, we acknowledged now that Nilsson's chronology is a rough outline, because his work predates the C-14 dating technique (Lines 239-240). However, if Nilsson's spruce pollen counts are from an Older Dryas layer as suggested in the study, we think it is correct to write “about 16-17 cal. yr BP” (Line 235). In fact, Reviewer #2 (see

comment 13 below) pointed out that that the pollen is not from the Older Dryas but, the Oldest Dryas.

Finally, we also agree with the reviewer that the sentence “Stable soils and warm summer temperatures could have been favourable for spruce to colonize the area” is general and relative. Now we explain better in the text that spruce, ecologically, is not a species requiring stable soils and high summer temperatures, as it is the treeline species in arctic Russia but can even grow in small pockets in the middle of the tundra and grows well on permafrost (Lines 246-249).

Comment #7: Line 236: The successful amplification of a DNA dating to around 43 ka suggests the possibility that such old sediments containing the here investigated markers were redeposited in the lake sediments resulting in the detection of Picea dating to 14.7 ka. Could you comment on this?

We understand this important point of concern of the reviewer and for a response please see our reply to comment #5.

Comment #8: Line 253: Nobody disagrees that Picea was present in the East - where in the East or how far north is a more interesting Question. Please be more specific.

Thank you for pointing this out. We added “, close to the ice sheet margins, possibly northwest Russia” to clarify better what we mean (Line 270).

Comment #9: Line 292 – 295: I know what you want to say here but readers less familiar with the debate may get lost here.

Thank you for pointing this out. We added a reference and rephrased the sentence, which now reads as follows (Lines 322-330): *Similarly, our results cannot exclude that some trees from the central Scandes originated from individuals surviving west of the SIS (either Andøya or the Trøndelag region)² however, our modern DNA analysis suggest that if they survived in the west, their genetic impact on the recolonization of Fennoscandia was negligible.*

Comment #10: Line 304: “via water transport”: It is not clear what is imagined here. - Seeds drifting on logs across the Baltic ice lake? But how to they get on top of the mountain via water transport?

Thank you for pointing this out. We rephrased the sentence to make it clearer (Lines 339-344). We are not assuming a single mean of long-distance transport, but multiple ones and we also acknowledge that these mechanisms are mostly not clear. We also cite Alsos *et al.* ‘Frequent Long-Distance Plant Colonization in the Changing Arctic ‘where the authors show that long-distance colonization of the Svalbard archipelago was the main and only means of postglacial recolonization in the Arctic and that long-distance events of different types have occurred repeatedly and from several source regions for many arctic species during the last glaciations.

Comment #11: Fig. 5: While you discarded the possibility of a survival of Picea south of the ice during the LGM in the text, the maps are still entertaining this possibility. So either you remove it from the maps or you discuss evidence for it in the text. I am not aware of any

evidence, but may of cause be wrong. Still these maps will be used in the same way you used the maps by Svenning et al.

As we have now tuned further down our conclusions and presented our scenarios as just hypotheses, we prefer to keep the southern spruce on the map. At the same time, to meet the reviewer's concern, we clearly state that this is a hypothesis and not a conclusion, suggested by both Nilsson and our data (Lines 514-515, 307-308). We also acknowledge that there is no strong and clear palaeoecological evidence supporting this (Line 319). Finally, we added a sentence on the legend of Figure 5 where we explain that, because we do not present solid palaeoecological evidence, we placed question marks on the trees in question.

Reviewer #2 (Remarks to the Author):

Comment #12: I am quite happy with the revision and recommend the paper for publication. As stated in my first review the new findings in the study is of large importance for the scientific field. A few minor things:

Row 134. Something is wrong with the sentence starting with "As the mh05..."

We thank the reviewer for pointing this out and we reformulated the sentence so that the meaning is clearer (Lines 140-143).

Comment #13: Row 220. Note that Nilsson's spruce pollen are from OLDEST Dryas (about 15,000-17,000 cal BP). And not from Older dryas. See reference 21.

We thank the review for spotting this mistake, we corrected it in the text (Line 235).

Comment #14: Row 255. Connection or not between the stem and the wood is important. I have a hard time (and lack of time) to scrutinize "The more in-depth answer" in the rebuttal, but from my perspective the reasoning seems convincing. Perhaps should it be included in the appendix?

Now we added this part in the supplementary material and add this information on line 288.

Reviewer #3 (Remarks to the Author):

Comment #15: Dear Authors,

Thank you for doing a very thorough revision of your manuscript. I think your replies are well argued.

We are happy the reviewer thinks our revision was thorough and we are grateful for the useful comments we received.

References

- Cheddadi, R., Vendramin, G. G., Litt, T., Francois, L., Kageyama, M., Lorentz, S., Laurent, J. M., de Beaulieu, J. L., Sadori, L., Jost, A. & Lunt, D. 2006: Imprints of glacial refugia in the modern genetic diversity of *Pinus sylvestris*. *Global Ecology and Biogeography* 15, 271-282.
- Hughes, A. L. C., Gyllencreutz, R., Lohne, O. S., Mangerud, J. & Svendsen, J. I. 2016: The last Eurasian ice sheets - a chronological database and time-slice reconstruction, DATED-1. *Boreas* 45, 1.
- Klaminder, J. unpublished data (see Figures 1 and 2 below) *
- Kullman, L. Boreal tree taxa in the central Scandes during the Late-Glacial: Implications for Late-Quaternary forest history. *Journal of Biogeography* 29(9), 1117–1124 (2002).. <https://doi.org/10.1046/j.1365-2699.2002.00743.x>
- Stroeven, A. P., Hattestrand, C., Kleman, J., Heyman, J., Fabel, D., Fredin, O., Goodfellow, B. W., Harbor, J. M., Jansen, J. D., Olsen, L., Caffee, M. W., Fink, D., Lundqvist, J., Rosqvist, G. C., Stromberg, B. & Jansson, K. N. 2016: Deglaciation of Fennoscandia. *Quaternary Science Reviews* 147, 91-121.
- Zale, R., Huang, Y. T., Bigler, C., Wood, J. R., Dalén, L., Wang, X. R., Segerström, U. & Klaminder, J. 2018: Growth of plants on the Late Weichselian ice-sheet during Greenland interstadial-1? *Quaternary Science Reviews* 185, 222-229

Figure 1. Within the city of Lycksele, we (Klaminder, unpublished data) excavated up to 14 m long Scots Pine trees (N=5) that grew on the site about 9.6-9.7 kyr cal BP according to conventional ¹⁴C dating of their wood. A) The excavation site shown with a red star in a

perspective of Fennoscandia. B) The excavation site as a function of the deglaciation line 10 kyr cal BP according to Hughes *et al.*, (2016). C) The excavation site as a function of the deglaciation chronology according to Stroeven *et al.*, 2016. According to pollen studies, the migration front of this tree species reached the tip of southern Sweden around 9 kyr cal. BP (Cheddadi *et al.*, 2006). These unquestionable findings of whole trees support our view that trees were able to rapidly (within a few hundred years) colonize deglaciated lands without being detected in pollen records.

Figure 2. Fire damage and wounds on a 9.6-thousand-year-old pine tree from Lycksele showing that there were not only pine trees, but a functional forest with burnable substrates just a few hundred years after deglaciation; (Klaminder, unpublished data).

Reviewers' Comments:

Reviewer #2:

Remarks to the Author:

Obviously, I cannot replace reviewer #1 as our background and expertise differ to a large extent. Anyway, I have checked the authors responses to reviewer #1's concerns and comments, and as far as I can judge they have made efforts to address them. Reviewer #1 concerns regarding the 14.7 date (Comment #1) is understandable but I think they have revised this section appropriately, and their statements are now rather cautious. sedDNA is novel and so are the results they present. But from that perspective the study deserves to be published as the authors now treat the data rather carefully. And importantly, they now give several convincing evidences contamination could be excluded. To some extent I agree with the reviewer as the result is spectacular and it could deserve a publication of its own. But that said, I sympathize with the authors putting it all in the same paper (except figure 5, see below), which actually makes it more interesting. It is also good they make it clear not using the most basal sediments (Comment #5). Regarding the reviewer's comment #11. Figure 5 is difficult to interpret and not really connected to the results and the text. And speculative. I suggest it should be omitted and published elsewhere when there is enough space to elaborate on the evidences at hand and the speculations.

I read through the manuscript again and as always, you find mistakes and question marks. See my comments below.

Row 33-34 (in the merged file): The first sentence in the abstract is important. I would prefer not the colon, please write it out.

Row 38: Write "Our other findings..." as you in the following sentence not include the southern Sweden finding.

Row 48: If picky Norway spruce also occurs in Belarus and the Baltic states.

Row 60: Should be "...limited number of pollen grains..."

Row 67: It is a theory that is problematic, not the direction of the establishment.

Row 86: Delete the word "other". It is not needed and makes it a bit confusing

Row 101: No need to mention it was suggested by Giesecke

Row 106: Delete "in".

Row 112: Not clear what the "first time" refers to. First time in Fennoscandia, or?

Row 120: As the reason for the interpretation is not stated here perhaps "as argued below" is needed.

Row 130: Delete one of the Lateglacial

Row 131: I suggest climates not to be in plural.

Row 135: Why is it "our" mitochondrial?

Row 140: Unclear which haplotype this refers to. It is moreover a long a complicated sentence which I do not fully understand.

Row 184: Delete the "and" after Regarding.

Row 192: Delete show.

Row 200: Not clear if older is referring to stratigraphy or studies.

Row 201. I am confused. Above you state that "In the central Scandes, spruce trees remained present throughout the Holocene". But here you say that "...we cannot draw conclusions regarding the continuous presence of spruce..."

Row 205: Why does a rare haplotype exclude contamination?

Row 218. Here Late Glacial, elsewhere Lateglacial. It should be consistent.

Row 221: Which of the two studies?

Row 225: Delete "also".

Row 239: Change from "the" to "these" to make clear it is referring to Nilsson's work. Besides, why ref 41 here?

Row 247: Use "environmental demands" instead of ecologically. And a reference for the statement is needed. As a matter of fact, spruce is sensitive to late spring frost, perhaps common during those times.

Row 248: What does middle refer to? Could be omitted.

Row 262: "Moreover" does not make sense here.

Row 282: That wave to Fennoscandia started before that, around 6000. See map in Seppä et al 2009.

Row 285: Unclear what "...the ones we sampled" refers to.

Row 286: I guess you mean "tree-lines".

Row 301: The south-eastern refers only to Finland and not Russia. Needs to be rephrases to make that clear.

Row 307: There are spruce macrofossil from Belarus from LGM, which is south of SIS.

- Binney, H.A. et al (2009) The distribution of late-Quaternary woody taxa in northern Eurasia: evidence from a new macrofossil database. Quaternary Science Reviews 28: 2445–2464.

Row 315: The last "and" should be reinserted.

Row 318. "From" instead of during. And Younger Dryas should be with uppercases.

Row 327: Could have been anywhere and not only where there happened to be paleo sites.

Row 328: Last sentence. Help the reader, why would the genetic impact be negligible?

Note: all line numbers revert to in the responses are based on the track and changes version of the manuscript.

Reviewer #2 (Remarks to the Author):

Obviously, I cannot replace reviewer #1 as our background and expertise differ to a large extent. Anyway, I have checked the authors responses to her concerns and comments, and as far as I can judge the they have made efforts to address them. Her concerns regarding the 14.7 date Comment #1) is understandable but I think they have revised this section appropriately, and their statements are now rather cautious. sedDNA is novel and so are the results they present. But from that perspective the study deserves to be published as the authors now treat the data rather carefully. And importantly, they now give several convincing evidences contamination could be excluded. To some extent I agree with the reviewer as the result is spectacular and it could deserve a publication of its own. But that said, I sympathize with the authors putting it all in the same paper (except figure 5, see below), which actually makes it more interesting. It is also good they make it clear not using the most basal sediments (Comment #5). Regarding the reviewer's comment #11. Figure 5 is difficult to interpret and not really connected to the results and the text. And speculative. I suggest it should be omitted and published elsewhere when there is enough space to elaborated on the evidences at hand and the speculations. I read through the manuscript again and as always, you find mistakes and question marks. See my comments below.

We removed figure 5 from the manuscript. We removed the reference to figure 5 in line 295, 303, 321.

Row 33-34 (in the merged file): The first sentence in the abstract is important. I would prefer not the colon, please write it out.

Line 33: We removed the colon.

Row 38: Write "Our other findings..." as you in the following sentence not include the southern Sweden finding.

Line 39: We modified the sentence.

Row 48: If picky Norway spruce also occurs in Belarus and the Baltic states.

Line 48: Yes, we should have been more accurate and we added now this information in the sentence.

Row 60: Should be "...limited number of pollen grains..."

Line 61: We corrected the sentence.

Row 67: It is a theory that is problematic, not the direction of the establishment.

Line 66: We agree and we corrected the sentence.

Row 86: Delete the word “other”. It is not needed and makes it a bit confusing

Line 86: ‘other’ was removed.

Row 101: No need to mention it was suggested by Giesecke

Line 101: “suggested by Giesecke (2013)” was removed.

Row 106: Delete “in”.

Line 106: “in” was removed

Row 112: Not clear what the "first time" refers to. First time in Fennoscandia, or?

Line 110: “first time” referred to that we were the first time to used both ancient sedimentary ancient DNA and molecular population genetics. However, this was true when we first submitted the text, but not any longer. We therefore removed “first time” to avoid both confusion and to be more accurate.

Row 120: As the reason for the interpretation is not stated here perhaps "as argued below" is needed.

Line 122: We added “as argued below” to the text.

Row 130: Delete one of the Lateglacial

Line 127: “During the Lateglacial” was removed

Row 131: I suggest climates not to be in plural.

Line 128: We agree and made climate singular.

Row 135: Why is it “our” mitochondrial?

Line 133: We think that this was because it was rephrased in a previous version, were we referred to our mitochondrial analysis. We changed it now to “The mitochondrial”

Row 140: Unclear which haplotype this refers to. It is moreover a long a complicated sentence which I do not fully understand.

Line 138-140: We have tried to make the sentence more comprehensible. The sentence meant to convey that, although the mh05 locus is present in most conifer taxa and, therefore, potentially also in Larix. Haplotype A (which is 20 base pairs shorter than haplotype B) is a rare mutation because it has only been observed in Norway spruce from Fennoscandia. It is therefore very unlikely that the haplotype A, that we found belonged to larch trees present in the area between 8.7-7.5 cal. kyr BP.

Row 184: Delete the “and” after Regarding.

Line 182: “and” was removed.

Row 191: Delete show.

Line 189: “show” was removed.

Row 200: Not clear if older is referring to stratigraphy or studies.

Line 197: We clarified that we meant the older studies.

Row 201. I am confused. Above you state that “In the central Scandes, spruce trees remained present throughout the Holocene”. But here you say that “...we cannot draw conclusions regarding the continuous presence of spruce...”

Line 199: we clarified that we meant here specifically southern Sweden. We only have samples between 15-10 cal. kyr BP and therefore we cannot draw conclusions about their presence in the Holocene.

Row 205: Why does a rare haplotype exclude contamination?

Line 203: We added “see supporting information for details”. The paragraph in the supporting information covering this point say.

“It is unlikely that haplotype A distribution was due to contamination during subsampling, because samples from the lakes in central Sweden were collected by different expeditions, and core opening occurred at different locations, and in areas where haplotype A was is absent or rare. It seems therefore very unlikely that contamination occurred at five different sites and exclusively at two specific age intervals (Table S6B). No positive PCR reactions were obtained in the two Russian sites, and at one site in southern Finland, haplotype A was recovered in two different PCR reactions. If contamination occurred, more positive reactions would have been observed at these sites.”

Row 218. Here Late Glacial, elsewhere Lateglacial. It should be consistent.

Line 42, 215, 245, 322: we change “Late Glacial” to “Lateglacial” in the text.

Row 221: Which of the two studies?

Line 218-219: We agreed that we were not clear enough and we modified the sentence by saying: The ancient DNA record however, had almost no chance of recovering spruce.

Row 225: Delete “also”.

Line 222: “also” was removed.

Row 239: Change from “the” to “these” to make clear it is referring to Nilsson’s work. Besides, why ref 41 here?

Line 235: We changed “the” to “these”. Ref 41, was not correct, we added a reference in the previous version and missed to change this number to 42 – we corrected this now.

Row 247: Use “environmental demands” instead of ecologically. And a reference for the statement is needed. As a matter of fact, spruce is sensitive to late spring frost, perhaps common during those times.

Line 242-243: We corrected this in the text and added a reference for this statement.

Row 248: What does middle refer to? Could be omitted.

Line 244: “We removed “the middle of”

Row 262: ”Moreover” does not make sense here.

Line 256: We agree and removed “Moreover” from the sentence.

Row 282: That wave to Fennoscandia started before that, around 6000. See map in Seppä et al 2009.

Line 271: It is true the wave starts around 6000, but we are here referring to the time spruce arrived to the Scandes. We clarified this in the sentence.

Row 285: Unclear what “...the ones we sampled” refers to.

Line 274: We refer here to the still living trees we sampled. We clarified this in the text.

Row 286: I guess you mean “tree-lines”.

Line 275: We corrected “tree-limit” to “tree-lines”

Row 301: The south-eastern refers only to Finland and not Russia. Needs to be rephrases to make that clear.

Line 291-292: We agree and we rephrased the sentence to make clear that the trees from south-east arrived from Russia earlier and before arriving to Dalarna.

Row 307: There are spruce macrofossil from Belarus from LGM, which is south of SIS.
• Binney, H.A. et al (2009) The distribution of late-Quaternary woody taxa in northern Eurasia: evidence from a new macrofossil database. Quaternary Science Reviews 28: 2445–2464.

Line 298-300. We added this to the manuscript, because the reviewer is right that Belarus is south of the SIS. We placed it earlier in the paragraph, because we already discussed that spruce trees likely didn’t arrive from the Baltic states. The same reasoning is true for The Belarus trees.

Line 354: we added “close to southern Sweden” to make sure we are clear we mean south of the SIS, and in near proximity to southern Sweden (this is now no longer possible to interpret from Figure 5 that was deleted)

Row 315: The last “and” should be reinserted.

Line 303: Corrected.

Row 318. “From” instead of during. And Younger Dryas should be with uppercases.

Line 306-307: Corrected.

Row 327: Could have been anywhere and not only where there happened to be paleo sites.

Line 314-315 We agree that it could have been anywhere, we therefore reformulated the sentence including such as, with reference 2, to indicate that these are the only places that have been analysed in the area.

Row 328: Last sentence. Help the reader, why would the genetic impact be negligible?

Line 317-318: We added a clarification to the sentence.